# Trivial improvements of predictive skill due to direct reconstruction of the global carbon cycle

Aaron Spring[1,2], István Dunkl[1,2], Hongmei Li[1], Victor Brovkin[1,3], and Tatiana Ilyina[1]

[1]Max Planck Institute for Meteorology, Hamburg, Germany
[2]International Max Planck Research School of Earth System Modelling, IMPRS, Hamburg, Germany
[3]Center for Earth System Research and Sustainability, University of Hamburg, Germany

**Correspondence:** Aaron Spring (aaron.spring@mpimet.mpg.de)

**Abstract.**

State-of-the art climate prediction systems include a carbon component recently. While physical state variables are assimilated in reconstruction simulations, land and ocean biogeochemical state variables adjust to the state acquired through this assimilation indirectly instead of being assimilated themselves. In the absence of comprehensive biogeochemical reanalysis products, such approach is pragmatic. Here we evaluate a potential advantage of having perfect carbon cycle observational products to be used for direct carbon cycle reconstruction.

Within an idealized perfect-model framework, we reconstruct a 50-year target period from a control simulation. We nudge variables from this target onto arbitrary initial conditions, mimicking an assimilation simulation generating initial conditions for hindcast experiments of prediction systems. Interested in the ability to reconstruct global atmospheric $CO_2$, we focus on the global carbon cycle reconstruction performance and predictive skill.

We find that indirect carbon cycle reconstruction through physical fields reproduces the target variations. While reproducing the large scale variations, nudging introduces systematic regional biases in the physical state variables, on which biogeochemical cycles react very sensitively. Initial conditions in the oceanic carbon cycle are sufficiently well reconstructed indirectly. Direct reconstruction slightly improves initial conditions. Indirect reconstruction of global terrestrial carbon cycle initial conditions are also sufficiently good reconstructed by the physics reconstruction only. Direct reconstruction improves air-land $CO_2$ flux negligibly. Atmospheric $CO_2$ is very well indirectly reconstructed. Direct reconstruction of the marine and terrestrial carbon cycles slightly improve reconstruction while establishing persistent biases. We find improvements in global carbon cycle predictive skill from direct reconstruction compared to indirect reconstruction. After correcting for mean bias, indirect and direct reconstruction both predict the target similarly well and only moderately worse than perfect initialization after the first lead year.

Our perfect-model study shows that indirect carbon cycle reconstruction yields satisfying initial conditions for global $CO_2$ flux and atmospheric $CO_2$. Direct carbon cycle reconstruction adds little improvements in the global carbon cycle, because imperfect reconstruction of the physical climate state impedes better biogeochemical reconstruction. These minor improvements in initial conditions yield little improvement in initialized perfect-model predictive skill. We label these minor improvements due to direct carbon cycle reconstruction *trivial*, as mean bias reduction yields similar improvements. As reconstruction biases

in real-world prediction systems are likely stronger, our results add confidence to the current practice of indirect reconstruction in carbon cycle prediction systems.

# 1 Introduction

Predicting variations in weather and climate yields numerous benefits for economic, social, and environmental decision-making (Merryfield et al., 2020). Carbon cycle prediction systems have the ability of predicting the near-term evolution of $CO_2$ fluxes (Li et al., 2019; Lovenduski et al., 2019a, b) and atmospheric $CO_2$ (Spring and Ilyina, 2020; Ilyina et al., 2021) to constrain the large internal variability of the global carbon cycle (Spring et al., 2020). Predictions require a forecasting model and initial conditions representing observations. However, due to sparse and temporally incomplete records, there is currently no global

biogeochemical reanalysis product to initialize Earth System Models (ESMs). Therefore, direct initialization of the carbon cycle, i.e. assimilating carbon cycle variables in ESMs, is not possible. State-of-the-art carbon prediction systems initialize the carbon cycle indirectly, by nudging the physical climate only, assuming that carbon cycle follows the initialized climate indirectly. However, this indirect carbon cycle initialization leaves the initial conditions of the carbon cycle unconstrained.

Here, we test how well indirect and direct carbon cycle reconstructions in an ESM initialize the carbon cycle in a perfect-model framework [Table 1 presents an overview which variables are reconstructed in which simulation]. We use the term reconstruction to describe methods of initialization of climate and the carbon cycle. Reconstructions aim to reproduce the evolution of the target, like a reanalysis product, in the ESM. Furthermore, we use the term "carbon cycle" to describe the processes exchanging carbon across the surface boundary between land, atmosphere and ocean, represented here by the air-

land and air-sea $CO_2$ fluxes. We ask the following research questions:

- How well can initial conditions be reconstructed in the global carbon cycle?

- Can initialization of the carbon cycle improve predictive skill of the carbon cycle?

In this perfect-model framework, we have perfect knowledge about the ground truth and a perfect model. Literally speaking, this asks how well could perfect observations be reconstructed in an ESM.


Originally, data assimilation is used to align the model state to an observations-based state, generally a reanalysis product (Schneider and Griffies, 1999; Meehl et al., 2009). However, here we use the same data assimilation technique to assess how well variables can be reconstructed in an idealized setup.

Thus, reconstruction in a climate model interferes with the freely running climate model yielding gains and drawbacks: The

main advantage of climate reconstruction is that the reconstruction forces the climate model to follow the target (Jeuken et al., 1996; Meehl et al., 2009). The main handicap associated with reconstruction is that the mass conservation is violated and that the model dynamics and feedbacks are obstructed (Zhu and Kumar, 2018). Consequently, circulation fields may change, and this has severe consequences for the biogeochemical tracer distributions in the ocean and carbon pools on land, because they are so sensitive and adapted to the previous climate state (Toggweiler et al., 1989). Therefore, reconstructions often lead to

biases. A partial solution can be bias removal by post-processing, which is feasible if the bias does not change the climate or ecosystem regime all together. Another solution is omitting nudging in regions strongly biased by reconstruction such as the

tropics, as demonstrated by (Park et al., 2018). Even if biogeochemical reanalysis products were available, it is unclear whether the reconstruction benefits correct these handicaps.

The lack of reanalysis products available for the reconstruction of carbon cycle initial conditions is often assumed as a weakness of the current predictions systems (Li et al., 2016; Séférian et al., 2018; Lovenduski et al., 2019b, a; Li et al., 2019; Ilyina et al., 2021), but to our knowledge an elaborate assessment is missing. The literature presents two alternative approaches to test the quality of reconstructed initial conditions:

In a perfect-model study, Servonnat et al. (2015) nudge only ocean surface temperature, salinity and sea-ice and assess

how well this surface reconstruction penetrates into the subsurface ocean physics, without addressing biogeochemistry in their analysis. This target reconstruction approach allows us to directly assess the quality of reconstructed initial conditions, which is useful and practical to know for forecaster issuing a forecast. Luo et al. (2017) use an equivalent simulation design, so called observing system simulation experiments (OSSEs), in which they assimilate sea surface temperature, sea surface salinity and sea surface height.

In a recent study Fransner et al. (2020) ask whether the initial conditions of ocean biogeochemistry or the initial conditions of ocean physics have a stronger influence on multi-year predictions using perfect-model twin perturbed initial conditions experiments. In the first set of hindcasts, they take identical initial conditions of the ocean physics to ensure identical climate evolution but completely different states from different members for ocean biogeochemistry. In the other set of hindcasts, they slightly perturb the ocean physics to force members on differing climate evolutions while keeping the ocean biogeochemistry

initial conditions identical. They find that ocean biogeochemistry initial conditions did not affect predictive skill later than the first lead year. Their approach asks the more theoretical question whether initial conditions of ocean biogeochemistry matter compared to ocean physics initial conditions.

We go beyond previous studies by using the methodology of Servonnat et al. (2015), with the aim to understand the quality of initial conditions reconstruction. In contrast to Fransner et al. (2020), we aim to answer the questions about quality of initial

conditions produced by different reanalysis approaches. We expand the scope by addressing the global carbon cycle, including the land, ocean and atmospheric compartments and the interactive exchange of $CO_2$ fluxes between them. We then assess the influence of these previously reconstructed carbon cycle initial conditions for initialized predictions of the natural carbon sinks and atmospheric $CO_2$. We focus on the global carbon cycle, because the land and ocean carbon cycle control the internal variability of atmospheric $CO_2$ (Friedlingstein et al., 2020).


After explaining the approach of target reconstruction in section 2, we separate reconstruction and its implication on predictive skill in two parts: We first evaluate reconstruction performance. We start with the physical reconstruction in section 3.1. Then we show how the ocean and land carbon cycles are reconstructed indirectly and how direct reconstruction can improve initialization in sections 3.2 and 3.3. We analyze the combined effects of the ocean and land reconstruction in the atmosphere

in section 4. Finally, the main findings and conclusions of this study are summarized in section 5.

## 2 Methods

### 2.1 Model Description

We use the Max Planck Institute ESM (Mauritsen et al., 2019, MPI-ESM), which was also used in the Coupled Model Inter-comparison Project Phase 6 (CMIP6) framework (Eyring et al., 2016). We run the model MPI-ESM1-2-LR, the low resolution configuration with 63 spherical harmonics in the atmosphere and with a horizontal resolution of about $1.8°$ on land, and about $1.5°$ in the ocean with daily coupling of the compartments. The time steps of the atmosphere/land and the ocean are 600 and 4320 s, respectively. We run the model with prognostic atmospheric $CO_2$ mixing ratio under pre-industrial conditions (*esm-piControl*).

The marine biogeochemical cycle model HAMOCC (Ilyina et al., 2013) is embedded in the ocean general circulation model MPIOM (Jungclaus et al., 2013). HAMOCC includes carbonate chemistry and an extended NPZD-type cycle including nutrient-light-temperature co-limitation and nitrogen-fixating cyanobacteria (Paulsen et al., 2017). The land carbon cycle model JSBACH includes dynamic vegetation, wildfires, soil carbon decomposition and storage (Schneck et al., 2013). The atmospheric general circulation model ECHAM6 transports the three-dimensional atmospheric prognostic atmospheric $CO_2$ tracer with a flux-form semi-Lagrangian scheme (Lin and Rood, 1996; Stevens et al., 2013).

### 2.2 Perfect-Model Target Reconstruction Framework

Simulations in a perfect-model target reconstruction framework aim to reproduce the target climate evolution (Griffies and Bryan, 1997; Servonnat et al., 2015), but are started from an independent initial state. Therefore the initial conditions of the reconstruction simulation and the the target to not match. But both target and initial conditions share the same climatology. We choose a 50-year target period from model years 1850 to 1900 and an uncorrelated restart file from model year 2005 from the pre-industrial control simulation (esm-piControl) submitted for the MPI-ESM1-2-LR model for C4MIP (Jones et al., 2016) in CMIP6 (Eyring et al., 2016).

In order to assess how many variables are needed to sufficiently reconstruct climate and biogeochemical cycles, we first perform reconstruction simulations only reconstructing physical state variables in atmosphere and/or ocean [Table 1]. In these simulations, the carbon cycle is only indirectly affected by the reconstruction of physical variables. In further simulations, we test how much carbon cycle states improve with respect to the target when carbon cycle state variables are reconstructed directly.

Newtonian or Haney (1974) relaxation, which is often called *nudging*, is a simple four-dimensional assimilation technique that dynamically reconstructs variables in an ESM. A non-physical relaxation term with relaxation coefficient $R$ (units $1/s$) is added to the prognostic equation to drag the model variable $X$, which is subject to model forcing $F_m$, towards its target $X_t$:

$$\frac{\delta X}{\delta t} = F_m(X) + R(X_t - X) \tag{1}$$

| | Reconstructed variables for each realm (nudging relaxation time-scale) | | | | |
|---|---|---|---|---|---|
| Reconstruction simulations | Atmosphere: temperature (24h) surface pressure (24h) vorticity (6h) divergence (48h) | Ocean (60d): temperature salinity | Sea-ice (60d): concentration thickness | Ocean carbon (60d): DIC alkalinity | Land: all JSBACH (reset restart files Jan 1st) |
| indirect$_{ATM\ only}$ | x | | | | |
| indirect$_{OCEAN\ only}$ | | x | | | |
| indirect | x | x | x | | |
| direct | x | x | x | x | x |

**Table 1.** Overview over different reconstruction simulations. The first column title marks the labels of the experiments as used in the manuscript. The reconstruction strength as relaxation time-scales is noted in brackets, where $h$ denotes hours and $d$ days. The land carbon cycle is not dynamically reconstructed at each time step, but by a hard reset of restart files each January 1st from the target run. These land restart files include carbon and nitrogen pools, soil physics (moisture, temperature, snow cover), vegetation cover (plant functional types distribution), and canopy (leaf area index).

For reconstruction of the dynamics of the ocean, we reconstruct three-dimensional temperature and salinity as well as sea-ice concentration and thickness [Table 1]. We label this reconstruction indirect [Table 1] from the carbon cycle's perspective, as the carbon cycle is not reconstructed directly, but instead indirectly follows the reconstructed physical climate. Observational ocean data is often not available at each model time step. Therefore, we interpolate (without adjustments preserving the temporal mean) monthly model target output to daily frequency as done in previous studies (Pohlmann et al., 2009). We choose a 60-day ocean relaxation time (converted to units $1/s$) like Servonnat et al. (2015) in their perfect-model target reconstruction study. Reconstructions towards observations usually choose a stronger nudging strength (Pohlmann et al., 2009; Keenlyside et al., 2008).

We reconstruct the physics of the atmosphere by nudging temperature, vorticity, divergence and the logarithm of surface pressure (Pohlmann et al., 2019). The high-frequency 6 hourly output serves as the target and is nudged into all 63 spherical harmonics. Temperature and the logarithm of surface pressure are nudged with a relaxation timescale of 24 hours, vorticity is nudged with a relaxation timescale of 6 hours, and divergence is nudged with a relaxation timescale of 48 hours. Relaxation coefficients are converted to units $1/s$ and are taken from previously used setups (Rast et al., 2012; Pohlmann et al., 2019; Li et al., 2019). Nudging the atmosphere with these quite short relaxation times is similar to the forced simulations, such as the Model Intercomparison Projects for ocean (OMIP) (Griffies et al., 2016; Orr et al., 2017), land (LMIP) (van den Hurk et al., 2016) and Global Carbon Budget (Friedlingstein et al., 2019) simulations, where (atmospheric) external boundary forcing drives the carbon cycle.

For reconstructions of oceanic carbon cycle, we use the same nudging approach and strength as for physical ocean reconstruction but on different variables. To reconstruct the components of the carbonate system, we nudge three-dimensional dissolved inorganic carbon (DIC) and total alkalinity [Table 1].

    Unfortunately, there is no nudging module available in the land surface model JSBACH. The current structure of JSBACH code is not flexible enough to allow frequent rewriting of physical variable fields, such as soil moisture or temperature, with

external data. Here, we choose to manually reset the initial conditions every January $1^{st}$ to the target values instead of the dynamic reconstruction at each time step. We thereby reconstruct land biogeochemistry and land surface physics such as soil moisture by resetting all restart variables every year. In supplementary information section D, we provide several sensitivity analyses by resetting land only every two or five years and resetting the ocean every year in the same way.

We compare the target with reconstructions in the various metrics showing different attributes of tracking performance: bias, anomaly correlation coefficient and root-mean-square-error. The non-physical relaxation terms in the prognostic equations can disturb the dynamics in the ESM and introduce biases defined as the differences in the reconstruction compared to the freely running target over time. The anomaly correlation coefficient skill score (ACC) shows the linear association between the reconstruction and the target over time and therefore measures synchronous evolution while ignoring bias. The root-mean-

square-error (RMSE) takes into account bias and measures the second-order euclidian distance between reconstruction and target simulation over time. Under the assumption that persistent biases can be removed by post-processing, we also assess RMSE after having the mean monthly bias removed. For equations please consult the supplementary [sec. A]. We calculate tracking performance over running 10-year chunks to capture the variability within tracking performance and reduce the influence of drifts over time.

How do we evaluate that a reconstruction is good enough? While good enough is a subjective judgement, we resample the target simulation along the time dimension with a block length of ten years to check the metric of two randomly compared 10-year chunks. We consider the $95^{th}$ quantile threshold for ACC and $5^{th}$ quantile threshold for the remaining distance-based metrics as a baseline of internal variability to be a good enough reconstruction (Efron and Tibshirani, 1993), which we will refer to as "resampling threshold" in the following.

## 2.3   Perfect-Model Predictive Skill Framework

In the second part of this study, we perform initialized perfect-model experiments (as in Spring and Ilyina, 2020). The simulations in the perfect-model framework are started from the indirect and direct reconstructions as well the target representing perfect initial conditions. We take 19 initialization states chosen every second January $1^{st}$ between 1860 and 1896, after allowing a 10 years adjustment phase after reconstructions were started. From each of those states from different reconstruction

simulations, we fork five ensemble members and simulate three lead years. The perfectly initialized ensembles are started from the target initial conditions without any previous reconstruction simulation. We generate ensemble members by perturbing the stratospheric horizontal diffusion by a factor of 1.0000{member} in the first year, e.g., the factor is 1.0005 for the 5th ensemble

member. This member generating approach provokes only tiny initial perturbations to the climate system as the ocean and land initial conditions remain identical.

We compute predictive skill as the root-mean-square-error (RMSE) between the ensemble mean and the target as verification (Wilks, 2006; Jolliffe and Stephenson, 2011) [Appendix A]. Please find additional details about the predictive skill metrics and the uninitialized bootstrapping in Spring and Ilyina (2020). Acknowledging that our reconstruction simulation developed biases and that biases are commonly reduced by post-processing in predictability research, we also apply a simple lead-time dependent mean bias reduction to the initialized ensembles to show whether skill improvements go beyond what a simple

post-processing could deliver. For each initialization in turns, we first calculate the mean bias for all but that given initialization and then remove that mean bias from the given initialization. This implies using information about future initializations as in bias-reduced hindcasts (Marotzke et al., 2016). We also evaluate predictive skill from a perfectly initialized ensemble, which are started from the perfect initial conditions taken from target simulation, whereas the ensembles from reconstructed initial conditions are biased with respect to the target [Fig. 5]. This initialized predictive skill is also compared with uninitialized

ensembles randomly generated from the target simulation representing ensembles without common initialization and hence no memory. This uninitialized reference skill is used in predictability research community to assign whether the skill increase stems from initialization.

## 3   Reconstruction in an Earth-System-Model

As the carbon cycle is sensitive to the climate evolution, we first assess how well the physical climate is reconstructed. Therefore, we first evaluate the physical climate state after reconstruction in subsection 3.1]. Afterwards, we assess how these different reconstructions of physical climate indirectly reconstruct the ocean, land and atmospheric carbon cycle in subsections and how direct reconstruction could improve initial conditions in subsections 3.2, 3.3 and 3.4].

### 3.1   Reconstruction of Physical Climate

Reconstructing the ocean and/or the atmosphere systematically disturbs the freely evolving model, which leads to annual mean biases with respect to the original target. We identify atmospheric circulation represented by winds and resulting precipitation and temperature to be descriptive for the impact of circulation on the carbon cycle. The gray stippling in figure 1 shows where this reconstruction bias is larger than the randomly resampling $5^{th}$ percentile mean absolute error threshold and therefore labeling the reconstruction not significantly better than internal variability.

All reconstructions yield identical results for winds and precipitation tracking performance. Reconstructing the ocean and/or the atmosphere introduces biases of up to 0.6 m/s in zonal and 0.9 m/s in meridional 10-m wind speed, depicting a southward shift of the Intertropical Convergence Zone (ITCZ). This bias results in a significant weakening of the equator-ward latitudinal winds, whereas extra-tropical latitudinal winds intensify [Fig. 1a]. The intensification and equator-ward shift of the easterly trade winds and weakening of the southern hemisphere westerlies are both not significant [Fig. 1b]. Precipitation is heavily

impacted by these biases in atmospheric transport across many regions of the globe. Precipitation significantly shifts southward at the equator with changes of more than 1 mm/day and increases in Western Canada, Western Russia and Southern Australia [Fig. 1c]. Unlike the previously described variables, the 2m-temperature bias depends on whether the ocean is reconstructed or not. Just reconstructing the ocean temperature and salinity (indirect$_{OCEAN only}$) leads to small, negative and significant biases in the tropical Atlantic and West Pacific. Also Northern and Southern Africa as well as the Amazon and China are subject to a small cold bias, whereas Saharan Africa and Southeast Asia gets substantially warmer. The polar regions cool significantly [Fig. 1d]. Only reconstructing the atmosphere (indirect$_{ATM only}$) leads to a warm bias nearly across the all oceans, but less cold bias over Northern and Southern Africa as well as China [Fig. 1e]. Combining atmosphere and ocean reconstruction (indirect) reduces the overall temperature bias, especially over the oceans [Fig. 1f].

While the above explained biases are liabilities of reconstructions, the linear association measured by the Anomaly Correlation Coefficient (ACC) benefits from reconstruction. Reconstruction recreates climate variability of the target [Fig. 1g-l]. The running 10-year correlation between the target and the reconstruction in atmospheric variables is in most grid cells above 0.4 and significantly better than the randomly resampling threshold. Reconstruction over the oceans is more successful in the tropics than in the extra-tropics, where the Northern and Southern Hemisphere mid-latitude westerlies have low, but still significant correlation. Generally, the atmosphere above the ocean is better reconstructed than above land, showing the stabilizing effect of an internally consistent ocean reconstruction on the atmosphere [Fig. 1g-l]. The Southern Hemisphere tropical convergence of winds is well reconstructed, but the meridional winds in central Canada and tropical Africa are not significantly reconstructed [Fig. 1g]. Also zonal winds across North America, Southern Africa and Siberia have low correlation with the target, but the tropical zonal winds are very well reconstructed [Fig. 1h]. Precipitation from the central Atlantic over central Africa is worse reconstructed than the resampling threshold, and the extratropical westerlies have low correlation with the target [Fig. 1i]. Temperature is well reconstructed in the tropical oceans [Fig. 1j-l]. Reconstructing both atmosphere and ocean (indirect) improves 2m temperature correlation better than only reconstructing a single realm. The indirect carbon cycle reconstruction is significantly better than the resampling threshold except in central Africa, where the ITCZ shift changes the climate regime [Fig. 1l].

This physical bias due to reconstruction, especially in the tropics, can be explained by the sensitivity of atmosphere-ocean coupling to perturbation induced by nudging (Milinski et al., 2016). Additionally nudging sea surface height might improve the ENSO thermocline feedback (Luo et al., 2017). The reconstruction of ocean and atmospheric variables is perfectly aligned with the model climatology into that same model. Hence, the reconstruction error does not arise from inconsistent observations, but from the perturbed interaction of atmospheric and oceanic dynamics. While reconstructing an increasing set of variables shows that nudging can be an efficient way to reconstruct variability (Jeuken et al., 1996), this reconstruction is biasing the climate state in the tropics at the same time (also explained in Zhu and Kumar, 2018).

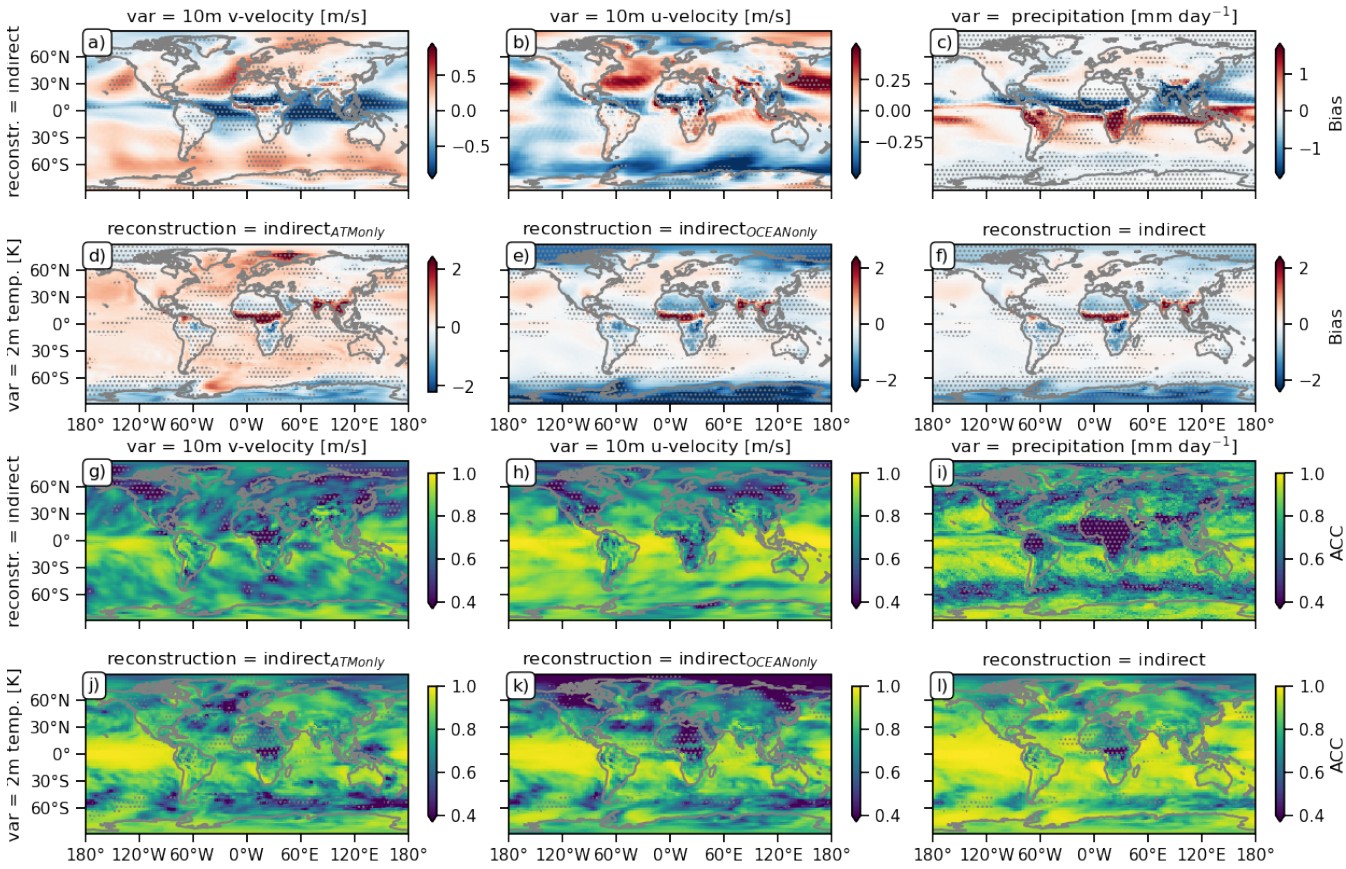

**Figure 1.** Spatial distribution of the bias (construction - target) (a-f) and anomaly correlation coefficient (ACC) (g-l) of different indirect carbon cycle reconstructions relative to the target over 10-year running windows of annual means [see Appendix A]. The reconstruction metrics for 2m temperature are shown for the indirect$_{ATM\ only}$ (d,j), indirect$_{OCEAN\ only}$ (e,k) and indirect reconstruction (f,l). Because of identical reconstruction skill for all indirect methods, only one indirect reconstruction is shown for other variables, zonal westward 10m wind (a,g), and meridional northward 10m wind (b,h), and precipitation (c,i). Gray stippling shows where the metric exceeds the 5$^{th}$ (for a-f) or 95$^{th}$ (for g-l) percentile threshold from random target block resampling, i.e. the reconstruction is not significantly better than internal variability.

Nudging atmospheric and ocean dynamics including sea-ice all at once (indirect reconstruction), as is often done in state-of-the-art carbon cycle prediction systems, brings large-scale improvements over random resampling and atmosphere-only (indirect$_{ATM\ only}$) reconstruction, but strong regional biases remain [Fig. 1].

### 3.2 Reconstruction of the Oceanic Carbon Cycle

How do these regional physical biases affect the reconstruction of oceanic carbon cycle? In order to assess the tracking performance in the indirect reconstruction of the oceanic carbon cycle, we focus on air-sea $CO_2$ flux and surface oceanic $pCO_2$ as the state variable of the ocean carbon sink, which is the oceanic driver of air-sea $CO_2$ flux (Lovenduski et al., 2019b).

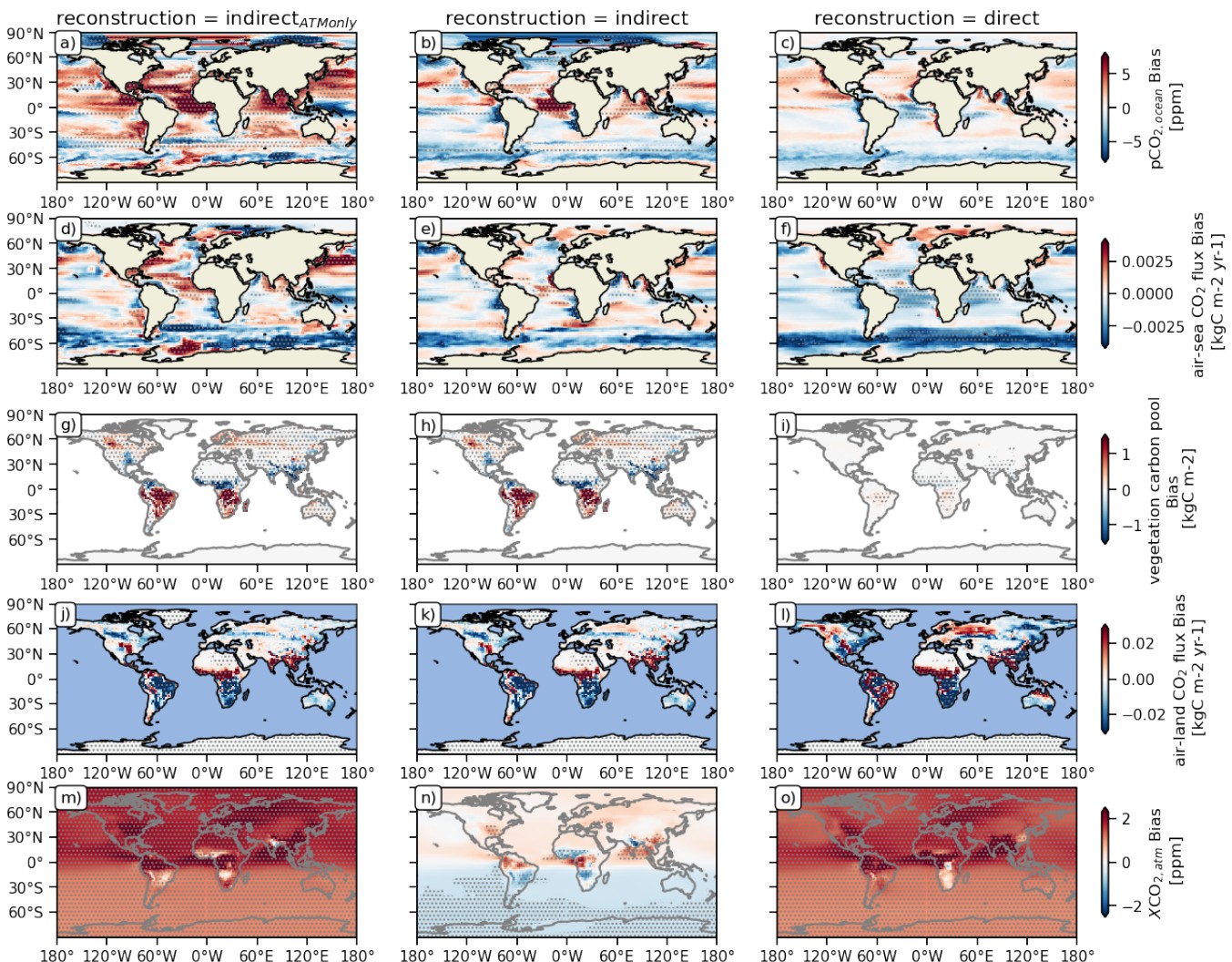

**Figure 2.** Spatial distribution of the bias between the target and different indirect carbon cycle reconstruction methods over 10-year running windows of annual means [see Appendix A]. Columns show the different carbon cycle reconstruction methods [see Table1]. Rows show the different variables: the ocean carbon cycle is represented by (a-c) the partial pressure of surface $CO_2$ in the ocean ($pCO_2$) and (d-f) surface air-sea $CO_2$ flux (negative values indicate carbon uptake by the ocean); the land carbon cycle is represented by (g-i) the vegetation carbon pools and (j-l) air-land surface $CO_2$ flux (negative values indicate carbon uptake by land); and the atmospheric carbon is represented by (m-o) the atmospheric $CO_2$ mixing ratio ($XCO_2$). Gray stippling shows where the bias exceeds the 5[th] percentile mean absolute error threshold from random target block resampling, i.e. the reconstruction is not significantly better than internal variability.

Reconstructing only the atmospheric dynamics (indirect$_{ATM only}$) leads to strong positive biases across large parts of the global ocean, which can be reduced by also reconstructing oceanic temperature and salinity (indirect) [Fig. 2a,b,d,e]. The weakening of the Southern hemisphere westerly winds decreases the magnitude of air-sea $CO_2$ flux, but more importantly

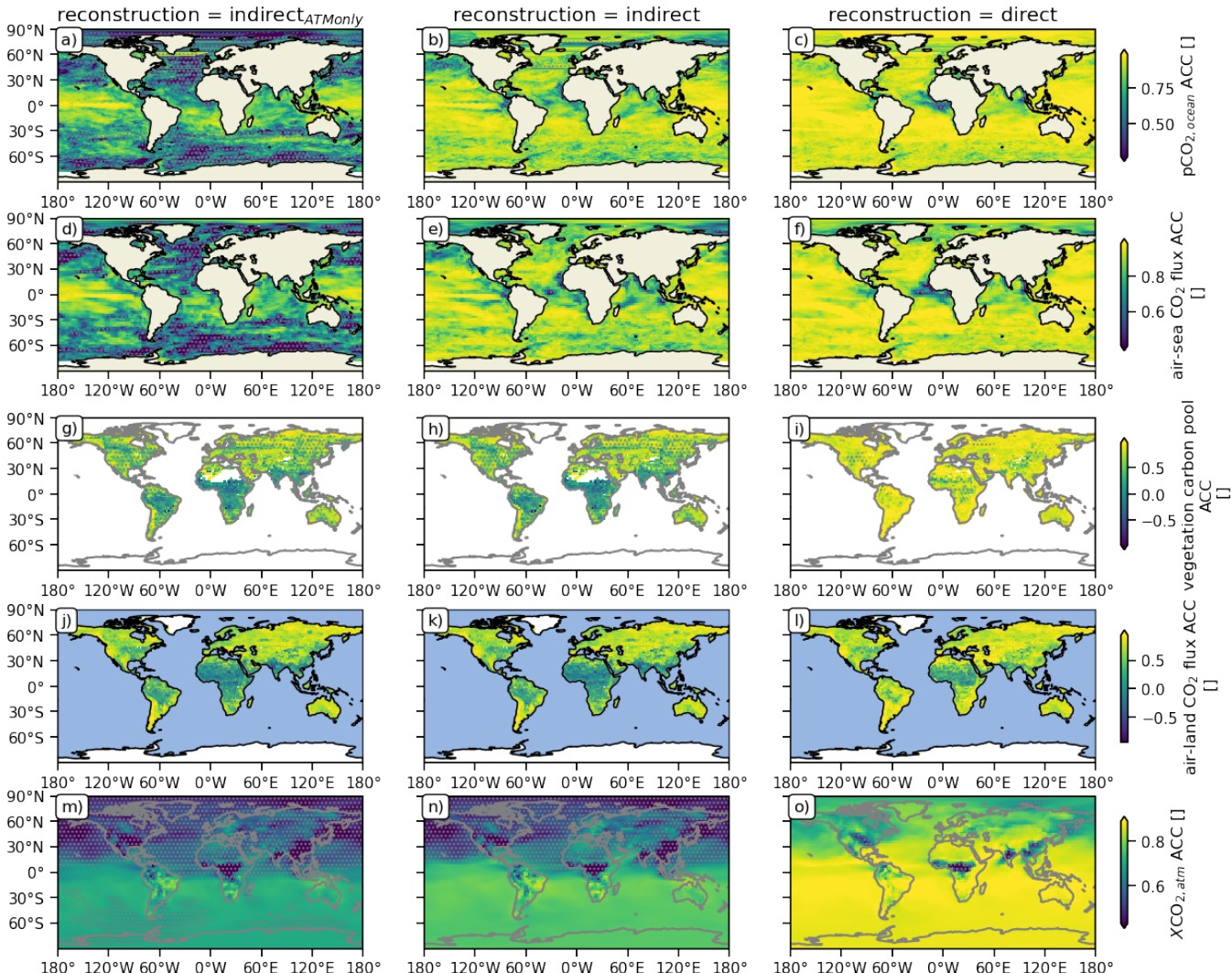

**Figure 3.** As Fig. 2 but for the anomaly correlation coefficient (ACC). Gray stippling shows where the ACC is lower than the 95[th] percentile ACC threshold from random target block resampling, i.e. the reconstruction is not significantly better than a resampling internal variability threshold.

reduces the Southern hemisphere overturning circulation and upwelling of carbon-rich waters, which leads to increased Southern Ocean carbon uptake [Fig. 2b,e]. The intensification of easterly trade winds [Fig. 1b] strengthens upwelling and therefore higher $pCO_2$ in the tropical Atlantic [Fig. 2b] (Lefèvre et al., 2013). The bias pattern of air-sea $CO_2$ flux is dominated by the bias of $pCO_2$ (Lovenduski et al., 2019b) [Fig. 2b,e].

The variations in the oceanic carbon cycle, described by the correlation coefficient, are better reconstructed than the resampling threshold. Indirect reconstruction of oceanic and atmospheric dynamics greatly improves tracking performance over

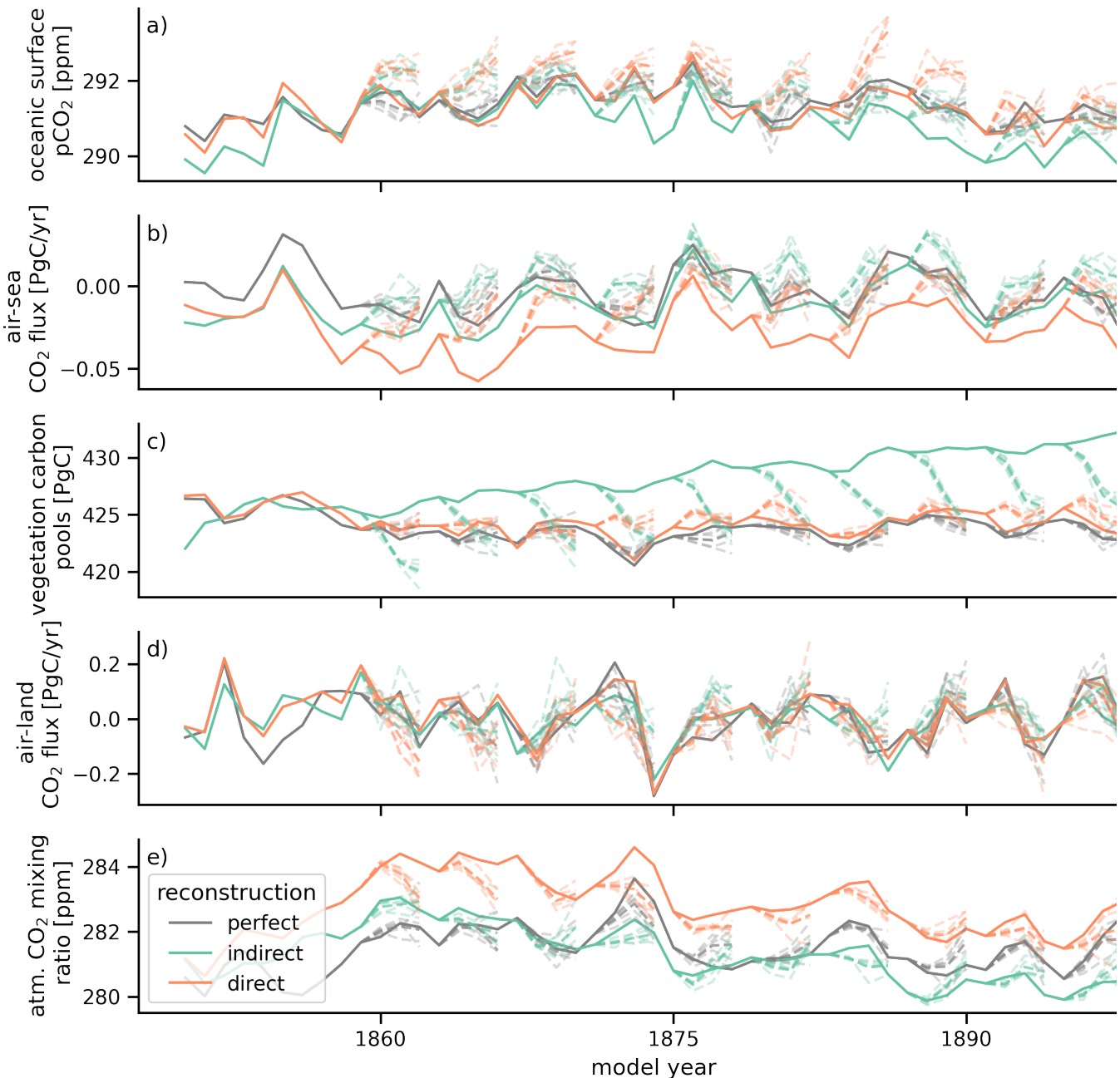

**Figure 4.** Evolution in global annual mean of (a) surface ocean pCO₂ (b), air-sea surface CO₂ flux (negative values indicate carbon uptake by the ocean) (c), vegetation carbon pools (g-i), air-land surface CO₂ flux (negative values indicate carbon uptake by land) (d) and atmospheric CO₂ mixing ratio (e). The target (gray) is quite well tracked by the indirect (green) and direct (orange) carbon cycle reconstruction. The solid line shows the different reconstruction simulations, the dashed lines show the initialized ensembles started from the different reconstructions.

atmosphere-only indirect$_{\text{ATM only}}$ reconstruction. The additional reconstruction of the physical ocean [Fig. 1e,f] enables largely

a correlation above 0.7 [Fig. 3b,e]. Only the carbon cycle in the tropical oceans remain difficult to reconstruct due to the strong biases in atmospheric circulation [Fig. 1a,b,c]. Note that the land and atmospheric carbon bias due to indirect reconstruction are discussed in subfigures 2g-o in sections 3.3 and 3.4).

Next, we compare the previously shown indirect carbon cycle reconstruction with direct carbon cycle reconstruction by

nudging dissolved inorganic carbon (DIC) and alkalinity (ALK) towards the target.

While direct oceanic carbon cycle reconstruction reduces the magnitudes of the bias across the ocean, biases are still evident [Fig. 2c,f]. These biases are caused by the physical biases, which the dynamical oceanic carbon cycle model is sensitive to. Hence, the biased ocean physics inhibits additional improvements in tracking performance from direct ocean carbon reconstruction.

Direct oceanic carbon cycle reconstruction improves the already high correlations across the oceans [Fig. 3c,f]. The resampling threshold is surpassed nearly everywhere. Only coastal areas, especially in the Eastern tropical Atlantic with strong wind and precipitation biases, have a correlation below 0.7.

Section 3.2 shows how well indirect and direct reconstruction of the ocean carbon cycle work overall. While the direct

reconstruction has slightly larger biases in air-sea $CO_2$ flux, direct reconstruction also brings higher correlation. Note that the land and atmospheric carbon bias due to direct reconstruction are discussed in subfigures 2g-o in sections 3.3 and 3.4).

### 3.3    Reconstruction of the Land Carbon Cycle

How do these regional physical biases affect the reconstruction of the land carbon cycle? In order to assess the tracking performance in the best indirect reconstruction of the land carbon cycle, we focus on the state variable cVeg, which represents

carbon storage in vegetation (leaves, stems, roots) and drives air-land $CO_2$ flux and hence the land carbon sink.

For the land carbon cycle, the reconstruction of the ocean temperature and salinity did not matter, when atmospheric temperature was also reconstructed [Figs. 2, 3]. Indirect reconstruction leads to biases compared to the target in carbon storage, and in particular cVeg [Fig. 2g,h], as the land carbon cycle is very sensitive to changes in atmospheric circulation, which are strongest in the tropics due to the ITCZ shift. In the Amazon and Southern Africa, the air-land $CO_2$ bias increases, most likely

caused by the strong positive precipitation bias in these regions [Fig. 1c; 2j,k]. Conversely, the carbon sink in Southeast Asia and central Africa has a carbon release bias due to less precipitation and a warm bias [Fig. 2j,k].

The reconstruction correlations in the land carbon cycle are much lower than for the oceanic carbon cycle. cVeg is well reconstructed in the extratropics, but the biases in the tropics result in correlations with the target lower than the resampling threshold [Fig. 3g,h]. Air-land $CO_2$ shows the same patterns with lower correlations, which are below the resampling threshold

in the tropics [Fig. 3j,k].

Direct reconstruction of the land carbon cycle, which is here performed by resetting all restart files of the land carbon sub-model to the target every Jan 1$^{st}$, greatly enhances tracking performance of cVeg by simulation design. A sensitivity analysis for less frequent resetting can be found in the supplementary information [section D].


This direct resetting reconstructs cVeg much better than the resampling threshold in the extra-tropics. However, the physical climate biases during the course of a year even introduce cVeg biases stronger than the resampling threshold in the tropics [Fig. 2i]. Also, the biases in the air-land $CO_2$ flux are not improved [Fig. 2l], which indicates that this hard reset of restart files introduces a shock to the dynamical land model.

On the other hand, correlations in cVeg and air-land $CO_2$ flux increased to above 0.5 everywhere expect in the tropics, where the ITCZ shift changes the climate regime [Fig. 3i,l].

Section 3.3 shows the direct land carbon cycle reconstruction yields stronger correlation improvements than ocean direct carbon cycle reconstruction, because the indirect reconstruction of the ocean was already quite good. Direct reconstruction

reduces biases in land carbon cycle state variables, but the resulting air-land $CO_2$ flux biases becomes worse.

### 3.4   Reconstruction of the Global Carbon Cycle and Atmospheric $CO_2$

Tracking performance for prognostic atmospheric $CO_2$ integrates the air-sea and air-land $CO_2$ fluxes over time (Spring and Ilyina, 2020; Spring et al., 2020). As atmospheric $CO_2$ mixes fast across the globe, we first examine globally aggregated quantities driving globally averaged atmospheric $CO_2$ [Fig. 4].

We first examine the indirect reconstruction represented by the green error bars in figures 5 and C1. The indirect reconstruction has a negative bias in global $pCO_2$ in the annual mean [Figs. 4a, 5a]. This bias is slightly higher than the magnitude as the resampling mean absolute error threshold, which resembles the temporal standard deviation [Fig. 5a]. The global oceanic $CO_2$ flux is low biased but within the resampling threshold magnitude range [Figs. 4b, 5d].

On the other hand, the variations of the global oceanic carbon cycle measured by ACC are well reconstructed surpassing the

resampling threshold [Fig. 5b,e].

When biases are persistent, they can be reduced by a bias reduction procedure, which is often done when applying climate model output to a real-world application. After applying a simple mean bias reduction, RMSE is well below the resampling threshold [Fig. 5c,f].

The indirect reconstruction also leads to biases in the land carbon cycle [Fig. 4c,d]. Vegetation carbon pools (cVeg) have a strong positive bias much larger than the resampling threshold [Fig. 4c]. The bias of global air-land $CO_2$ flux is very small in the annual mean.

Global annual cVeg has a 0.5 correlation with the target, which is lower than the resampling threshold. Global air-land $CO_2$ variations are well reconstructed surpassing the resampling threshold [Fig. 5h,k].

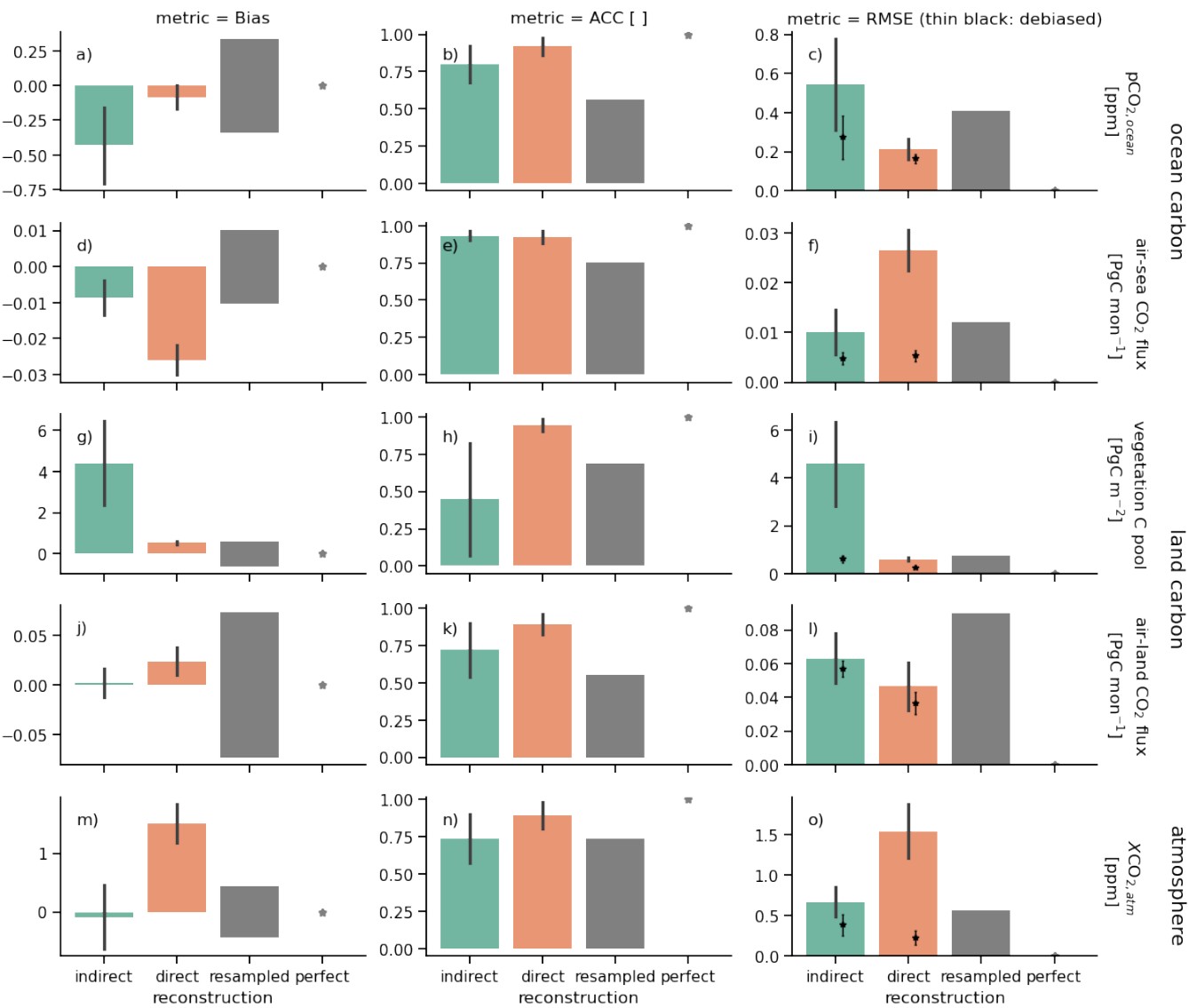

**Figure 5.** 10-year running mean annual reconstruction skill in bias (left), anomaly correlation coefficient (ACC, middle) and root-mean-square-error (RMSE, right) for global aggregation of carbon cycle variables: (a-c) surface oceanic partial pressure of $CO_2$, (d-f) air-sea $CO_2$ flux (negative values indicate carbon uptake by the ocean), (g-i) vegetation carbon pools, (j-l) air-land $CO_2$ flux (negative values indicate carbon uptake by land) and (m-o) mixing ratio of atmospheric $CO_2$. Errorbars show $\pm\sigma$ standard deviation of the running skill over time. Columns show different reconstruction methods: indirect (green) and direct (orange). The gray bar marks the magnitude of the 95[th] percentile for ACC and 5[th] percentile for bias and RMSE of a random reconstruction skill block-bootstrapped from the target control simulation as an unskillful reference. Gray stars indicate perfect skill. Thin black errorbars with crosses show RMSE skill after a mean bias reduction.

Without bias reduction, accuracy measured by RMSE is worse than the resampling cVeg threshold. After bias reduction, cVeg accuracy is still slightly worse than the threshold, but accuracy improves from 5 PgC to below 1 PgC, which is the magnitude of the resampling threshold [Fig. 5i] Global air-land $CO_2$ flux accuracy is below the resampling threshold [Fig. C1l].

Global atmospheric $CO_2$ has larger variations in reconstruction skill, depending on which 10-year chunk is used to calculate the metric. And the skill has a nearly constant level throughout the year [Fig. C1m-o]]. The mean bias is close to zero [Figs. 4e, 5m]. Correlation with the target is above 0.7 and in the range of the resampling threshold [Fig. C1n]. Accuracy is at 0.7 ppm in the range of the remsampling threshold. Mean bias reduction improves accuracy to below 0.5 ppm [Fig. 5o].

Understanding the tracking performance of the ocean and land carbon cycle, we can now evaluate the spatial distribution of globally averaged atmospheric $CO_2$. Reconstructing only the atmosphere warmed the globe and also increased atmospheric $CO_2$ globally [Figs. 1k, 2m]. Reconstructing additionally also the ocean keeps the temperature stable, but introduces a less than 1 ppm low bias across the Southern Hemisphere, reflecting the higher uptake of the Southern Ocean carbon sink and the Southern Hemisphere land carbon sink [Fig. 2e,k,n]. The variations in atmospheric $CO_2$ are well reconstructed with correlation coefficients above 0.6 in the Southern Hemisphere, but across the Northern extra-tropics and the land regions with strong physics biases correlation is at 0.5 below the resampling threshold [Fig. 2m,n].

Now, we assess the potential improvements in the global carbon cycle due to direct reconstruction of the global carbon cycle variables shown in orange in figures 5 and C1.

The global ocean carbon cycle improves after direct DIC and alkalinity reconstruction [Fig. 5a]. Monthly biases remain but are now within the resampling threshold [Fig. C1a]. Correlation improves from 0.8 to above 0.9 in surface $pCO_2$. Air-sea $CO_2$ correlation does not improve, but only because of correlations above 0.9 for the indirect reconstruction were already very high [Fig. 5b]. Correlation for boreal winter is above 0.95, indicating that initial conditions in winter are well reconstructable to initialize forecasts with for the oceanic carbon sink [Fig. C1b]. Direct reconstruction improves $pCO_2$ accuracy to 0.2 ppm. Mean bias reduction can hardly improve accuracy after direct reconstruction [Fig. 5c]. Air-sea $CO_2$ flux accuracy degrades in comparison to indirect reconstruction. This degradation is removed by the mean bias reduction [Fig. 5f].

All results for the direct reconstruction of the land carbon cycle must be understood in the context of the method chosen for the direct reconstruction: Because we reset the restart files in Jan 1$^{st}$ to the target, the metrics are near to perfect in January by design. However, then the biogeochemistry is not modified directly for twelve months and only follows the physical climate reconstruction indirectly, so biases triggered by physical biases unaligned with the reset land biogeochemistry pools quickly build up and may approach the metric of the indirect reconstruction. Likewise, there is no bias in global cVeg in January by design. The bias increases with the physical biases, until surpassing the resampling threshold in August increasing until the end of the year [Fig. C1g]. Annual cVeg bias is strongly improved by direct reconstruction [Fig. 5g]. Global air-land $CO_2$ flux has a stronger bias than the indirect reconstruction [Fig. 5j]. Correlation in the global cVeg is near perfect in January by design and slowly decreases to 0.8 in December while still better than the resampling threshold [Fig. C1h]. Annual cVeg variations are

much better reconstructed by the direct method compared to the indirect [Fig. 5h]. Global air-land $CO_2$ flux variations increase by 0.2 [Fig. 5k]. Direct reconstruction improves global cVeg accuracy. Accuracy is better than the resampling threshold after mean bias reduction. Direct reconstruction slightly improves $CO_2$ flux accuracy. Furthermore, a mean bias reduction slightly improves accuracy [Fig. 5i,l].

The global $CO_2$ bias in the direct reconstruction increases to +1.8 ppm [Fig. 5m], but correlation increases from 0.7 to 0.9 [Fig. C1n]. The direct reconstruction has worse accuracy than the indirect due to established bias, but after mean bias reduction the accuracy is below 0.3 ppm [Fig. 5o].

How does direct carbon cycle reconstruction affect tracking performance in prognostic atmospheric $CO_2$? Already the time series indicate, that there is a 1-2 ppm atmospheric $CO_2$ positive bias in the direct reconstruction [Fig. 4e]. This bias is very

homogeneous over the oceans [Fig. 2o]. However, correlation strongly increased to 0.9 above the oceans and above 0.7 on land except for central Africa with its persistent biases, where the reconstruction is not better than the resampling threshold.

Section 3.4 shows that atmospheric $CO_2$ follows the reconstructed land and ocean carbon cycle integrating their respective fluxes over time. The direct carbon cycle reconstruction introduces a large bias in the atmospheric $CO_2$ distribution that the

indirect reconstruction did not suffer from, even after mean bias reduction [Fig. B3]. Globally averaged atmospheric $CO_2$ after direct reconstruction had a better accuracy tracking performance after the mean bias reduction, showing how global aggregation can balance regional biases. The direct land and ocean carbon cycle reconstructions track target much better than the indirect reconstruction, when measured by correlation.

Hence, in large, this first part showed how direct carbon cycle reconstruction improves linear association between recon-

struction and target (measured by ACC), but often increases biases degrading accuracy (measured by RMSE). Only after bias reduction, accuracy improves with respect to the indirect carbon cycle reconstruction.

## 4  Impact of Reconstruction on Global Carbon Cycle Predictive Skill

The second part of the paper assesses how predictive skill improves due to direct initialization of global carbon cycle variables. Specifically, we verify the RMSE between the five ensemble members initialized from the indirect and direct reconstructions across all initializations based on raw and lead-time dependent bias corrected timeseries [Figs. 4, 6].

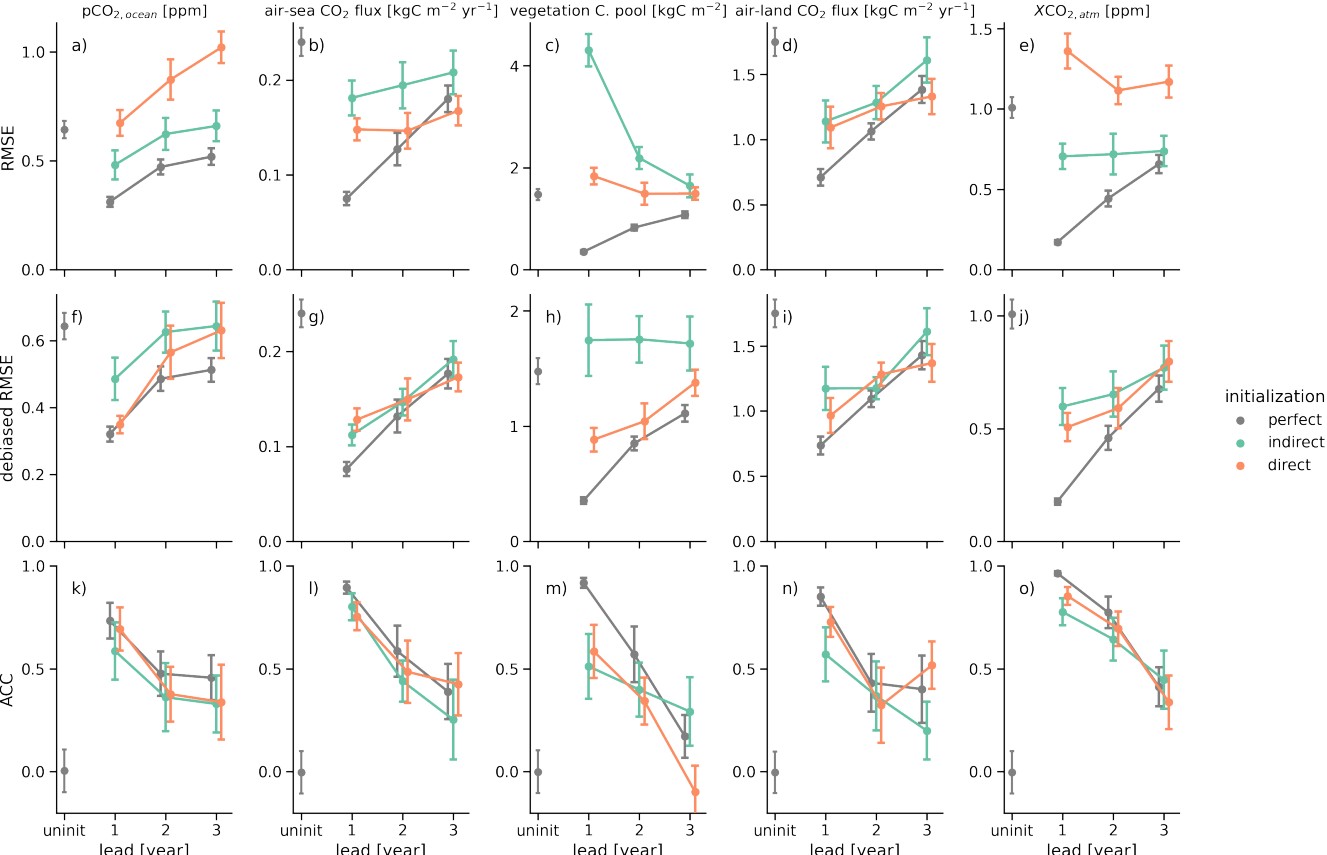

**Figure 6.** Predictive skill measured by (a-e) root-mean-square-error (RMSE), (f-j) RMSE after bias reduction and (k-o) anomaly correlation coefficient (ACC) between the initialized ensemble mean and the target as a function of lead year for different initialization setups: perfect indicating no reconstruction and hence perfect initial conditions to predict the target (gray), indirect (green) and direct (orange). Columns show global variables: for the ocean carbon cycle (a) oceanic surface $pCO_2$, (b) air-sea $CO_2$ flux; for the land carbon cycle (c) total land carbon pools, (d) air-land $CO_2$ flux and in the atmosphere (e) atmospheric $CO_2$ mixing ratio. Initialized ensembles are resampled with replacement (N=500) along the initialization dimension to account for initialization sampling uncertainty (see Spring and Ilyina, 2020), where errorbars show the resampled initialization skill uncertainty ($\pm 1\sigma$). Uninitialized ensembles, shown at lead 0, are resampled from the target control simulation and show the reference skill without initialization.

## 4.1 Oceanic Carbon Cycle

The RMSE between the initialized ensembles and the target simulations in annual globally-averaged $pCO_2$ continuously increases from lead year one to lead year three as expected. While perfectly and indirectly initialized ensembles stay below the resampling uninitialized threshold for the first two lead years indicating that global $pCO_2$ is predictable due to initialization [Fig. 6a], the direct initialization has a larger error due to the offsets in global atmospheric $CO_2$, which $pCO_2$ tries to equilibrate to [Fig. 4e]. Therefore this persistent bias causes lead year three to be not predictable. A simple mean bias reduction resolves this issue making all three lead years predictable. Direct initialization only beats indirect initialization for lead year one with RMSE of $0.35\pm0.05$ ppm versus $0.45\pm0.05$ ppm [Fig. 6f].

Global air-sea $CO_2$ flux is predictable for three years in all initialization methods, which is one year longer than in Spring and Ilyina (2020), possibly because here we use more and more equally distributed initialization dates. Direct initialization is advantageous over the indirect initialization, because the initial lead offset is smaller ($0.14\pm0.01$ PgC/year vs $0.18\pm0.02$ PgC/year) [Fig. 6b]. The simple mean bias reduction improves skill of the non-perfect initializations to identical magnitudes [Fig. 6g].

## 4.2 Land Carbon Cycle

Indirect initialization makes cVeg not predictable. The physical reconstruction biases drive larger errors in lead year one than in later lead years, also to a lesser extent for the direct reconstruction where some biases are corrected. But both reconstructed initialized ensembles show decreasing distances towards the target, whereas increasing distances are expected for vanishing predictive skill as in the perfectly initialized ensembles [Fig. 6c]. Mean bias reduction eliminates the differences between direct and perfect reconstruction making both predictable unlike the indirect reconstruction [Fig. 6h].

Global air-land $CO_2$ flux is predictable for three years, again one year longer than found in Spring and Ilyina (2020). Both reconstructed initializations start with a higher error of $1.1\pm0.2$ PgC/year in lead year one compared to perfect initialized $0.7\pm0.1$ PgC/year [Fig. 6d]. Mean bias reduction brings non-perfect initializations within the error bars of the perfect initialization after lead year one [Fig. 6i]. A recent analysis focused on process-based understanding of land carbon predictability using JSBACH indicates that soil moisture as well as soil carbon storage, both reconstructed by the direct method, influence the air-land $CO_2$ flux at most (Dunkl et al., 2021, under review).

## 4.3 Atmospheric $CO_2$

Perfect and indirect initialization atmospheric $CO_2$ predict the target for three years as found in Spring and Ilyina (2020). While the perfect initialization error grows continuously from zero, the indirect initialization error stays nearly constant at $0.7\pm0.1$ ppm, but the error stays below the direct initialization error, which suffers from the bias in the direct reconstruction simulation [Fig. 6e]. Mean bias reduction improves RMSE, making direct initialization better but still within the margins of the indirect initialization. After lead year one, indirect and direct initializations are similar to perfect-initialization predictive skill at 0.7

ppm [Fig. 6j].

The anomaly correlation coefficient measures how predictable variations are and is independent of the mean bias [Fig. 6k-o] (Jolliffe and Stephenson, 2011). Measuring predictive skill with ACC shows very similar behaviour across all variables. While perfect initialization is best predictable, indirect and direct carbon cycle initialization are fairly similar. Predictive ACC skill seems to saturate after lead year 2.

These initialized predictive skill results show that indirectly initialized ensembles predict the target quite reasonably. Direct initialization suffers strong shocks in some variables, when reconstruction is started and stopped, but these shocks can be partly reduced by a mean bias reduction. The improvements of direct reconstruction over indirect reconstruction in the global carbon cycle predictive skill after bias reduction are not significant, except for vegetation carbon pools (cVeg) [Fig. 6f-j].

## 5 Summary and Conclusions

In this study, we assess how well the global carbon cycle is reconstructed in an ESM and how well a ground truth target simulation can be predicted by these initializations.

The main limitation of land carbon cycle reconstruction potential is the hard reset of restart files which is fundamentally different to the dynamical nudging applied for ocean and atmospheric physics. Our study represents a first attempt to quantify whether initial conditions reconstruction in land carbon cycle is indeed needed for addressing predictive skill of the global carbon sinks and atmospheric $CO_2$ concentration. For a real-world application, our direct land carbon reconstruction method should not be used. In practice satellite products of carbon cycle variables could be assimilated into the model periodically or

at each time step. However, just strong interference with the model will likely result in strong drifts, especially in dependent variables. For useful real-world applications of land carbon cycle assimilation, sequential (Evensen, 1994; Balmaseda et al., 2007; Zhang et al., 2007) or variational (Han et al., 2004) data assimilation techniques could be used for initialization. But still the problem of data availability for the reforecast period remains. Haney reconstruction is the simplest approach to data assimilation allowing little flexibility to the model. Many centers are now transitioning towards the ensemble Kalman filter

data assimilation which allows more variability (Park et al., 2019; Brune and Baehr, 2020). Applying such techniques to the carbon cycle may lead to better reconstructions. A final limitation of the method is that we use a model to reconstruct to itself. Therefore we do not have any structural uncertainty other than the reconstruction method itself and no processes missing in our framework. When reconstructing the real world, our model lacks processes and resolution contributing to structural uncertainty.

     We find that reconstruction, which is an interference into the freely evolving model, leads to biases in physical climate.

Because of its sensitivity to physical climate, the global carbon cycle is heavily biased itself by these physical biases. In ESMs, first the atmosphere, then the ocean and only then the carbon cycle is equilibrated and tuned for pre-industrial control conditions. Once reconstruction slightly modifies the mean state in the physical climate, the sensitive carbon cycle deviates from the near-equilibrium state. A previous study reported biases after reconstruction (Zhu and Kumar, 2018). Yet, to our knowledge, we present the first attempt at reconstructing in a perfect-model framework, where no biases due to climatology

differences are expectable. Zhu and Kumar (2018) also mention that reconstruction ability likely depends on the model and application area, hence there seems to be no out-of-the-box solution for all ESMs. However, additionally nudging sea surface height might improve the ENSO thermocline feedback (Luo et al., 2017).

     We furthermore find that the commonly used indirect reconstruction of carbon cycle, in which only climate physics are reconstructed and the carbon cycle follows indirectly, tracks the target reasonably well. A resampling threshold corresponding

to internal variability is surpassed across large parts of the globe. Only the areas with strong physical and consequently carbon cycle biases miss that benchmark occasionally. For the ocean carbon cycle, the reconstruction of the physical ocean fields is critical to reconstruct carbon cycle initial conditions, which explain why current state-of-the-art carbon cycle prediction systems have skill despite not initializing the ocean carbon cycle with ocean carbon cycle observations (Séférian et al., 2014; Park et al., 2018; Li et al., 2019; Lovenduski et al., 2019b).

Direct reconstruction of ocean and land carbon cycle improves bias, association and accuracy on a grid cell level, but aggregated on the global scale, direct reconstruction does not improve over the indirect reconstruction significantly. Also after a mean bias reduction, which is a common post-processing technique applied to model output for real-word use, accuracy measured in RMSE after direct reconstruction is only slightly better, often still overlapping with indirect reconstruction. Because the advantage of direct reconstruction can similarly be achieved by a simple mean bias reduction, we label these direct reconstruction improvements *trivial* with respect to the indirect method on the global scale. More advanced data assimilation methods may yield better reconstruction skill for the carbon cycle (Han et al., 2004; Balmaseda et al., 2007; Zhang et al., 2007).

When the success of atmospheric $CO_2$ reconstruction is evaluated, caution is needed. Reconstruction of the ocean and land carbon sink can easily introduce offsets from the target, because reconstruction violates conservation of mass by creating or erasing carbon. This can easily lead to offsets in the sinks which quickly accumulate in atmospheric $CO_2$. If $CO_2$ reconstruction is the focus, i.e. in reconstructing the transient climate from $CO_2$ emission, and offsets appear, adjustments of atmospheric $CO_2$ might be needed to correct for these offsets. However, we find that these offset biases are only of the order of 1-2 ppm in a perfect-model framework, which is small compared to the range of carbon feedbacks seen in atmospheric $CO_2$ in transient simulations. Hence, these offsets due to the restart files are not in our focus. Rather, equilibrated land and ocean carbon sinks with reconstructed climate determine realistic reconstructed atmospheric $CO_2$.

In the second part, we find that predictive skill after indirect initialization is similarly good as after direct initialization. This means that oceanic carbon cycle initial conditions are much less important that physical ocean initial conditions for oceanic carbon cycle predictions, which confirms the findings of (Fransner et al., 2020). Reconstructed initialized predictive skill is close to perfectly initialized predictive skill after mean bias reduction, especially after lead year one.

Because the improved global predictive skill after direct reconstruction can similarly be achieved by a simple mean bias reduction and predictive skill after both reconstructions mostly overlaps, we label these direct reconstruction predictive skill improvements *trivial*, with respect to the indirect method on the global scale. This result is similar to Fransner et al. (2020), who find that ocean carbon cycle initial conditions matter much less than physical ocean initial conditions for annual carbon cycle predictions.

We conclude that the indirect carbon cycle reconstruction serves its purpose of reconstructing variation in the global carbon cycle. However, our study is designed and conducted in an idealized framework. When transferring our results into assimilation of real-world observations and its implications on predictability, structural uncertainties (model resolution in space and time) and missing ecosystem processes need to additionally be dealt with. Future studies, especially those aiming to address regional marine ecosystems, could consider a wider range of assimilation techniques and data breadth. Furthermore, more advanced data assimilation techniques (Evensen, 1994; Han et al., 2004; Balmaseda et al., 2007; Zhang et al., 2007) should be explored. Reducing the physical climate bias with its consequences for the carbon cycle holds more potential for improvements in initial conditions and predictive skill than direct carbon cycle initialization (Saito et al., 2011; Lee and Biasutti, 2014; Hua et al., 2019).

Nevertheless, our results add confidence to the current practice of indirect reconstruction in carbon cycle prediction systems (Ilyina et al., 2021).

495       We now provide a climatology figures G1 and G2 in the supplementary for context of the biases.

*Code and data availability.* Forecast verification was performed with the python package CLIMPRED (Brady and Spring, 2021) [https://github.com/pangeo-data/climpred/], which was co-developed with Riley X. Brady from University of Colorado, Boulder. Scripts and data to reproduce this analysis are archived in http://hdl.handle.net/21.11116/0000-0007-A697-3.

# Appendix A: Metrics

## A1 ACC

The anomaly correlation coefficient (ACC) assesses the synchronous evolution over time of the forecast, here reconstruction $x(t)$ and the reference, here target $\hat{x}(t)$, (Jolliffe and Stephenson, 2011) and is defined as:

$$ACC(x(t),\hat{x}(t)) = \frac{cov(x(t),\hat{x}(t))}{\sqrt{var(x(t)) \cdot var(\hat{x}(t))}} = \frac{\frac{1}{T}\sum_{t=1}^{T}(x(t) - \overline{x(t)})(\hat{x}(t) - \overline{\hat{x}(t)})}{\sqrt{\frac{\sum_{t=1}^{T}(x(t)-\overline{x(t)})^2}{T}} \cdot \sqrt{\frac{\sum_{t=1}^{T}(\hat{x}(t)-\overline{\hat{x}(t)})^2}{T}}}. \tag{A1}$$

## A2 RMSE

In the initial conditions reconstruction part, the root-mean-square-error (RMSE) measures the second-order distance between forecast $x(t)$, here reconstruction $x(t)$ and the reference, here target $\hat{x}(t)$, (Jolliffe and Stephenson, 2011) and is defined as:

$$RMSE(x(t),\hat{x}(t)) = \sqrt{\frac{\sum_{T=1}^{T}(x(t) - \hat{x}(t))^2}{T}}. \tag{A2}$$

As a predictability metric, the root-mean-square-error (RMSE) measures the second-order distance between forecast $x(t)$ and the target $\hat{x}(t)$ over lead time $t$ (Jolliffe and Stephenson, 2011). RMSE is calculated over all initialisations $N$ and every member $M$ is used as a forecast and verified against the target. RMSE is defined as:

$$RMSE(x(t),\hat{x}(t)) = \sqrt{\frac{\sum_{i,j=1}^{N,M}(x_{i,j}(t) - \hat{x}_j(t))^2}{NM}}. \tag{A3}$$

## A3 Bias

We set the target as the ground truth. Therefore any deviation from the reconstructions $x(t)$ to the target $\hat{x}(t)$ is seen as a bias, analogous to the bias between a model simulation (reconstruction) and observations (ground truth).

$$bias(t) = x(t) - \hat{x}(t) \tag{A4}$$

## A4 Removing the Bias

After removing the mean bias from reconstruction $\overline{x(t)}$ and target $\overline{\hat{x}(t)}$, the RMSE is also calculated as debiased RMSE.

$$RMSE_{\text{debiased}}(t) = RMSE\left((x(t) - \overline{x(t)}, \hat{x}(t) - \overline{\hat{x}(t)}\right) \tag{A5}$$

## A5 Running Metric

We calculate the mean tracking performance ($mtp$) over time for all metrics as a running mean over $s = 10$ years. This reflects that reconstructions are supposed to reconstruct the given climate states within months to a couple of years and the metric should not be prone to long-term trends that are not captured by the reconstruction. We ignore the first $c = 10$ years (out of

$t_{max} = 48$ years) of reconstruction, where the model experiences an initial shock after adjusting to the new reconstructed
climate (Kröger et al., 2017).

$$tpm(metric) = \frac{1}{t_{max} - s - c} \sum_{t=c}^{t_{max}-s} metric(x_{(t=t..t+s)}, \hat{x}_{(t=t..t+s)}) \tag{A6}$$

## A6  Resampling Threshold

To get an estimate of random tracking performance due to internal variability, i.e. how well one 10-year chunk tracks just
another random 10-year chunk, we randomly resample 10-year chunks from the target simulation and apply the same tracking
metrics. As a baseline skill from this random resampling in the figures, we take the 95% threshold for ACC and the 95% for
the remaining distance-based metrics to ensure that the tracking performance from a reconstruction simulation is only worse
compared to one out of 20 randomly resampled 10-year chunks.

## Appendix B: Reconstruction RMSE Maps

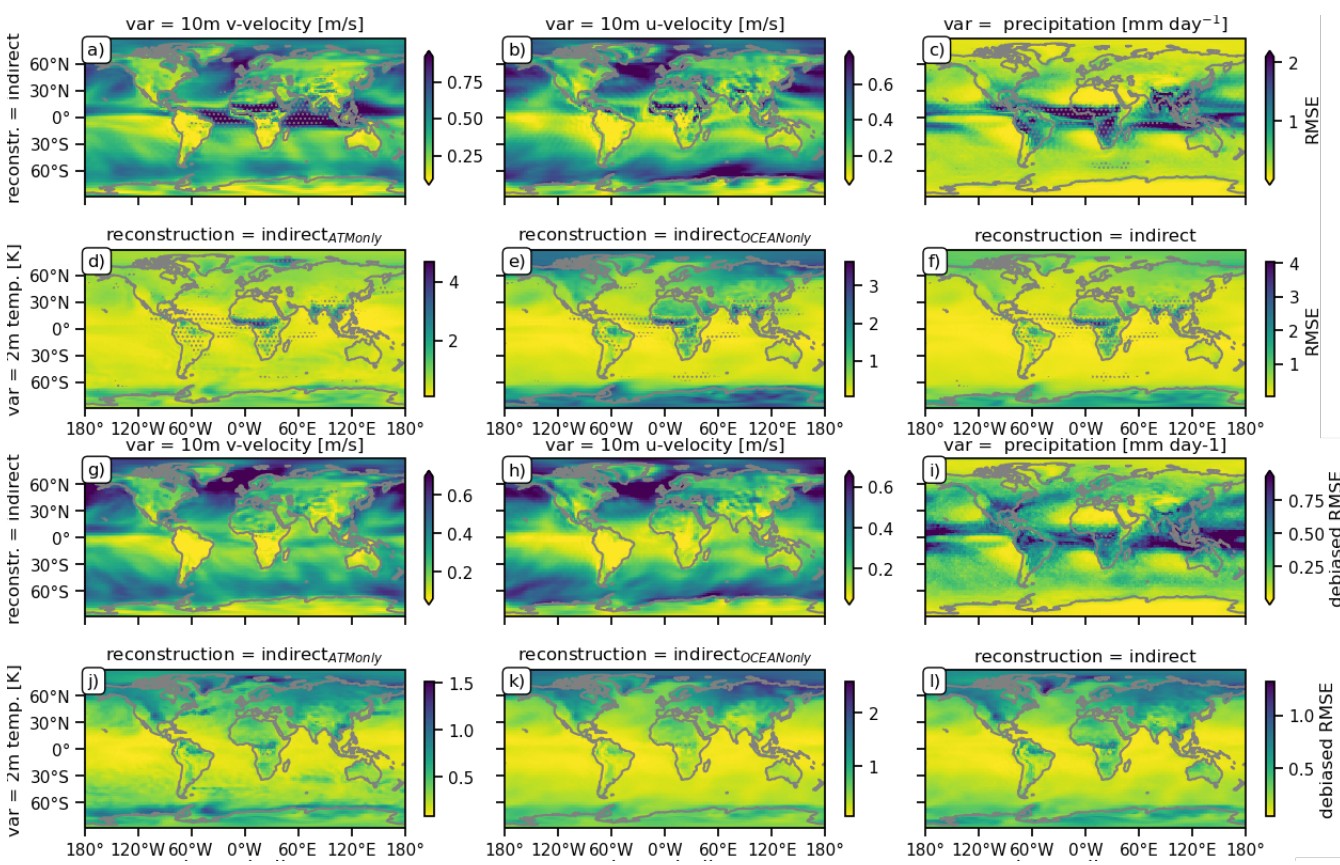

**Figure B1.** As Fig. 1 but for Root-mean-square-error (RMSE) (a-f) and for RMSE after bias reduction (g-l).

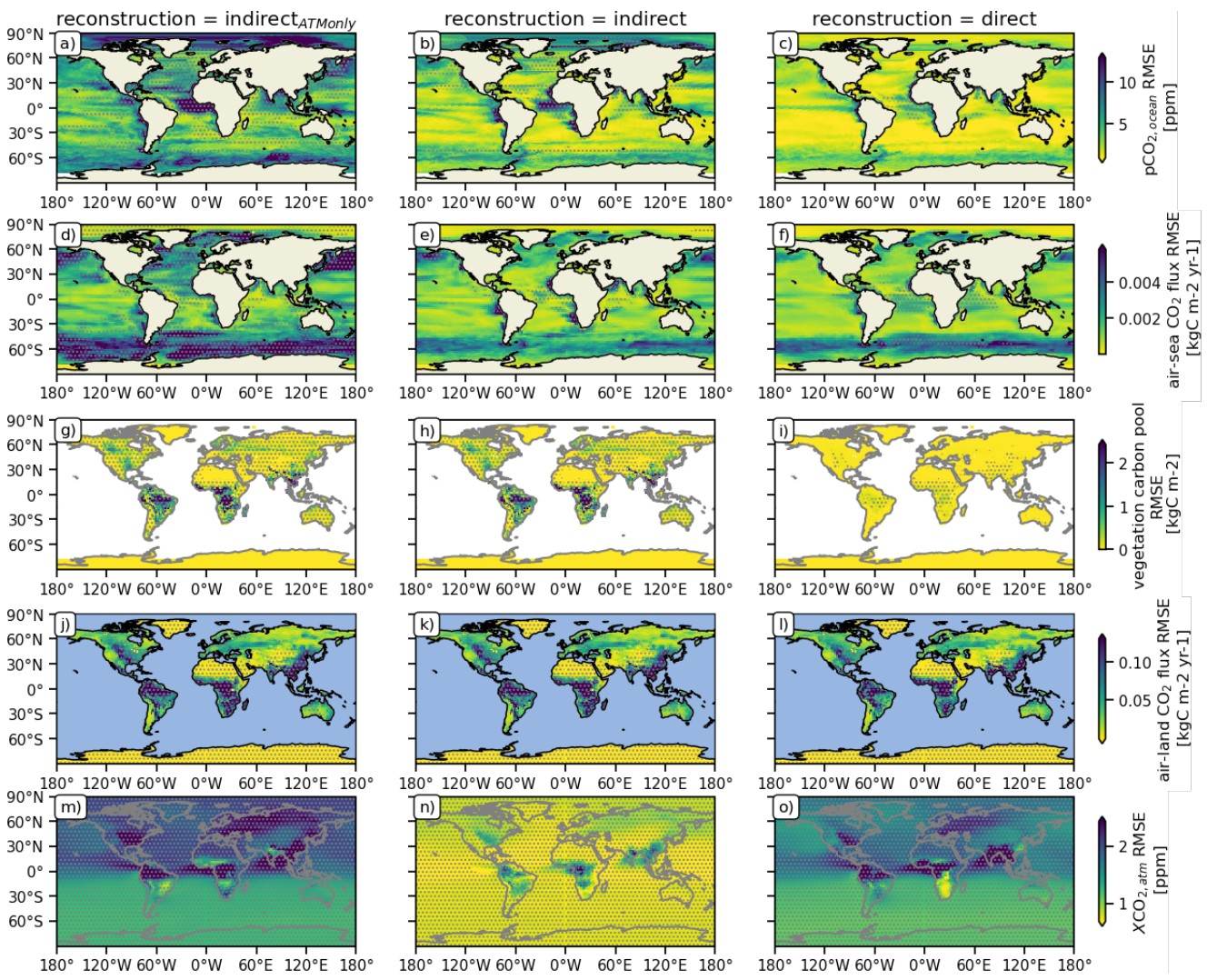

**Figure B2.** As Fig. 2 but for the Root-mean-square-error (RMSE). Gray stippling shows where the RMSE is worse than the 5th percentile RMSE threshold from random target block resampling, i.e. the reconstruction is not significantly better compared to internal variability.

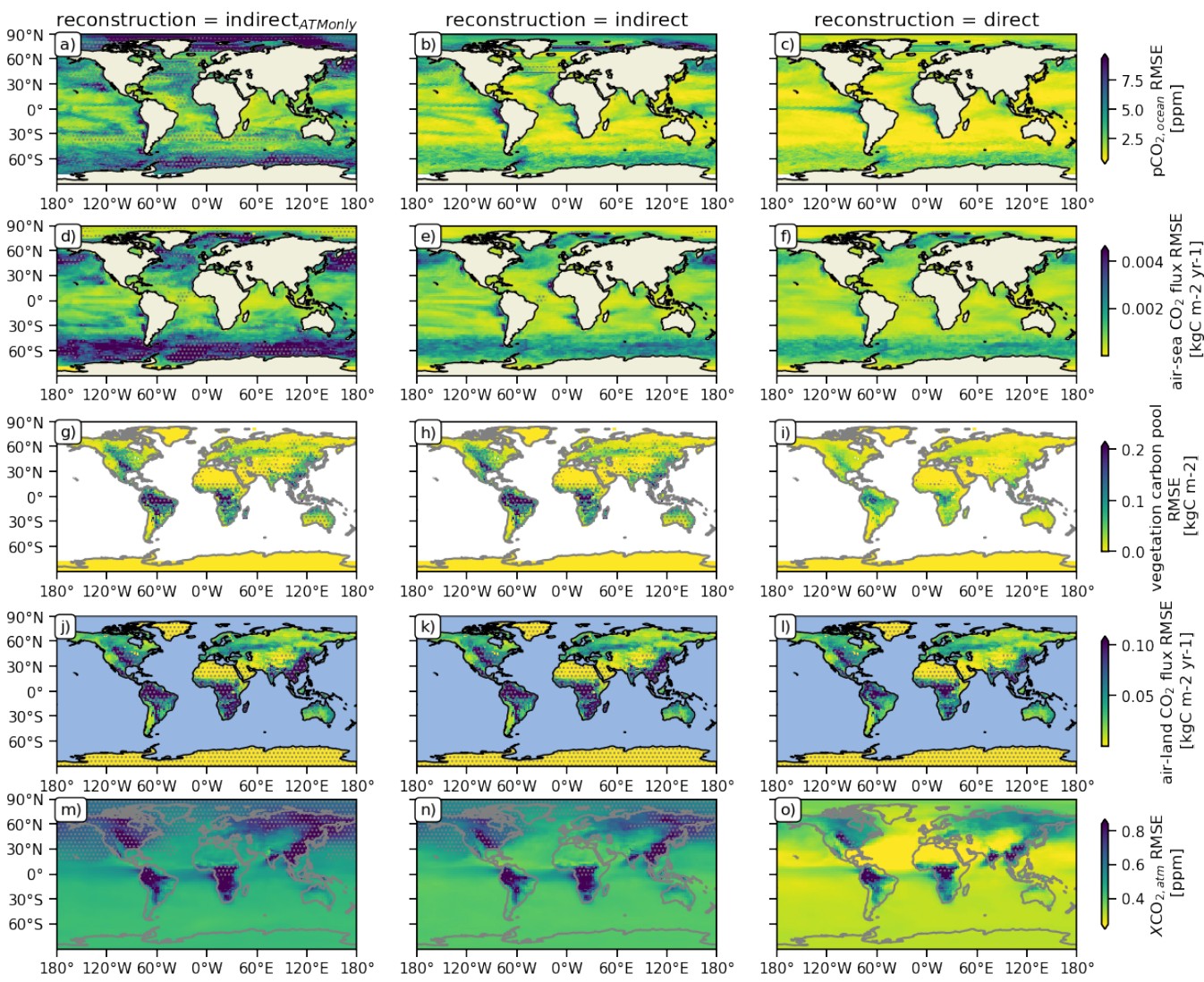

**Figure B3.** As Fig. B2 but for Root-mean-square-error (RMSE) after bias reduction.

## Appendix C: Monthly Global Tracking Performance

In order to explain the effect of the direct reconstruction in the land carbon cycle on global reconstruction performance, Fig.
         C1 shows the tracking performance on monthly timeseries, whereas Fig. 5 show only results for annual timeseries.

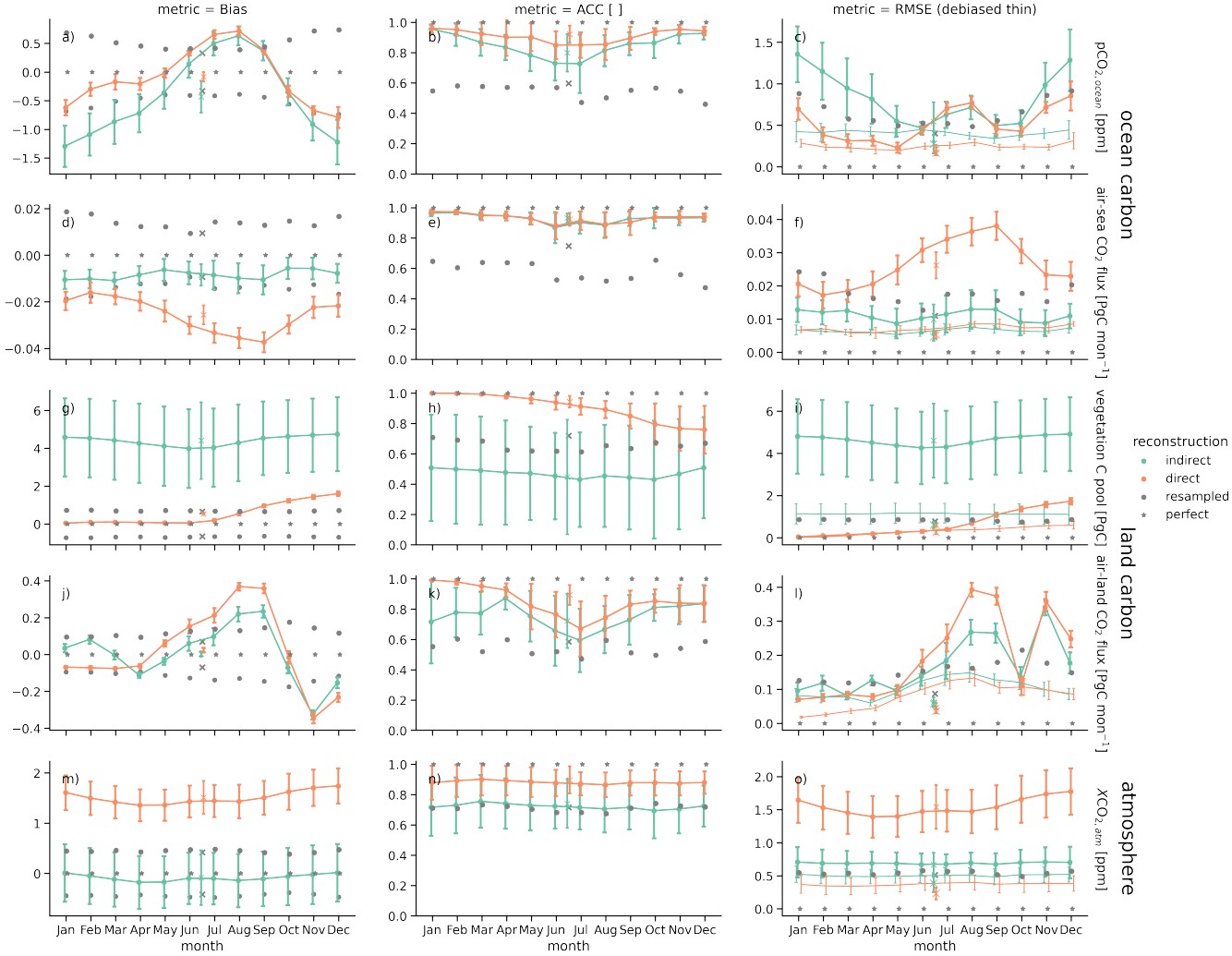

**Figure C1.** 10-year running mean reconstruction skill per month in bias (left), anomaly correlation coefficient (ACC, middle) and root-mean-square-error (RMSE, right) for global aggregation of carbon cycle variables: (a-c) surface oceanic partial pressure of $CO_2$, (d-f) air-sea $CO_2$ flux (negative values indicate carbon uptake by the ocean), (g-i) vegetation carbon pools, (j-l) air-land $CO_2$ flux (negative values indicate carbon uptake by land) and (m-o) mixing ratio of atmospheric $CO_2$. Whiskers show the 5th and 5th percentile of the running skill over time. Colors show different reconstruction methods: indirect (green) and direct (orange). Gray stars indicate perfect skill. Gray dots mark 95th percentile for ACC and 5th percentile for the remaining distance-based metrics of random reconstruction skill block-bootstrapped from the target control simulation as an unskillful reference skill. Crosses show reconstruction skill of annual mean timeseries. Thin lines show monthly RMSE skill after a mean bias reduction.

## Appendix D: Sensitivity Analysis for Different Reconstruction Timestep in ...

### D1  ... on Land Carbon Cycle

We perform sensitivity reconstructions of the land restart file resetting to understand how sensitive this reconstruction method
to the frequency of resetting. We performed additional simulations resetting the land model on Jan. 1$^{st}$ every second or every
fifth year [orange triangles in fig. D1].

Global cVeg starts by definition with perfect skill in Jan after a reset. When resetting only every second year, the mean
January tracking performance is already decreased, and decreases further. The negative correlations for five-year resetting
shows the shock to the system if not immediately balanced by further resetting in the every (second) year case.

The global air-land $CO_2$ flux correlation degrades for less frequent resetting towards the indirect performance, but bias and
accuracy improve.

Global atmospheric $CO_2$ aggregates these results and is also sensitive to biases developing in both sinks. Here, less frequent
resetting of the land carbon cycle reduces the bias and therefore accuracy.

The tracking accuracy is of similar magnitude after mean bias reduction.

### D2  ... on Ocean Carbon Cycle

We perform the same kind of restart file resetting reconstruction to the ocean model [blue line in fig. D1]. The motivation here
is to see whether a resetting of the ocean carbon cycle also yields perfect accuracy (RMSE) skill for January. But the ocean
carbon cycle is sensitive to the physical climate and hence the direct ocean carbon cycle resetting accuracy degrades compared
to the indirect tracking bias and accuracy, only correlation increases [Fig. D1a-f]. Contrary to resetting restart files in the land
model, initial conditions accuracy measured by RMSE does not approach perfect skill of 0, because the physical climate did
not experience this hard reset but is nudged dynamically.

In general, this hard reconstruction also seems to work for the ocean carbon cycle, because the tracking performances are
not very different from the indirect method [Fig. D1a-f].

The tracking accuracy is of similar magnitude after mean bias reduction.

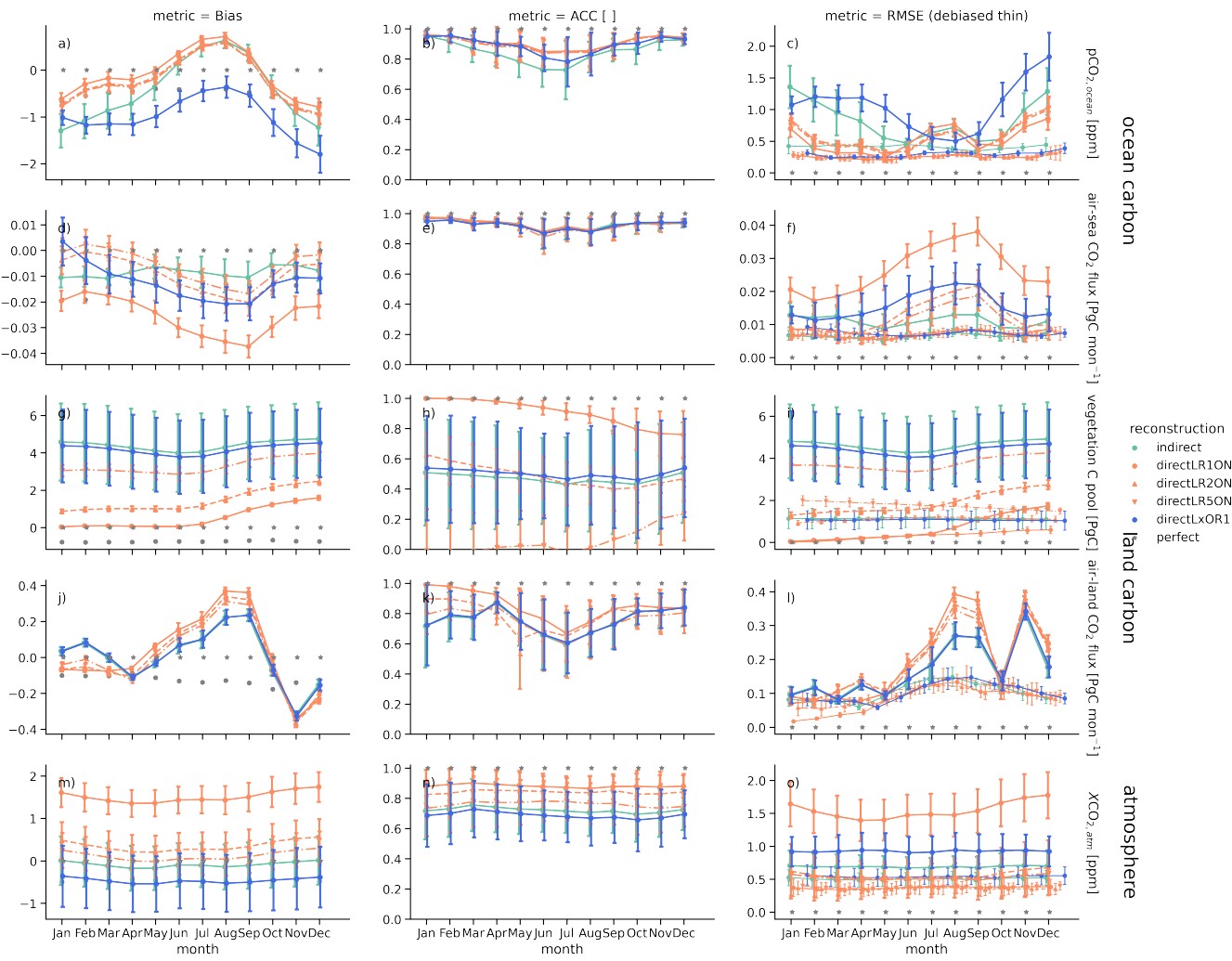

**Figure D1.** As Fig. C1 but for sensitivity simulations of the restart file resetting reconstruction. In all simulations the physical climate is nudged as in indirect [Table 1]. directLR1ON describes land resetting every year and ocean nudging and is the indirect simulation. directLR2ON describes land resetting every second year and ocean nudging. directLR5ON describes land resetting every fifth year and ocean nudging. directLxOR1 describes no land reconstruction and ocean setting every year.

 **Appendix E: Seasonality**

In reference for figure C1 to better understand reconstruction skill in context of target seasonality:

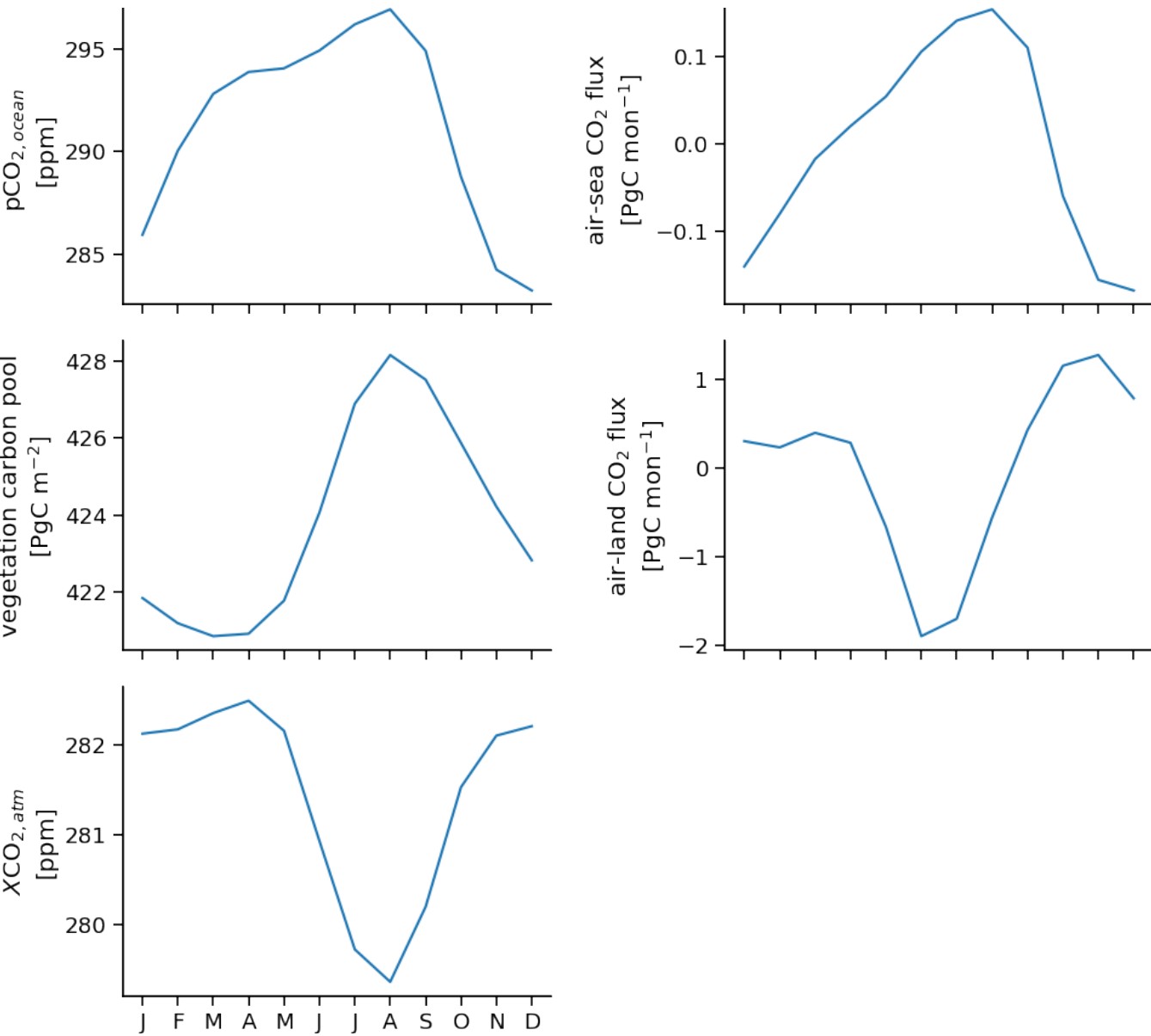

**Figure E1.** Seasonality of the target simulation for global aggregated carbon cycle variables.

## Appendix F: Schematics

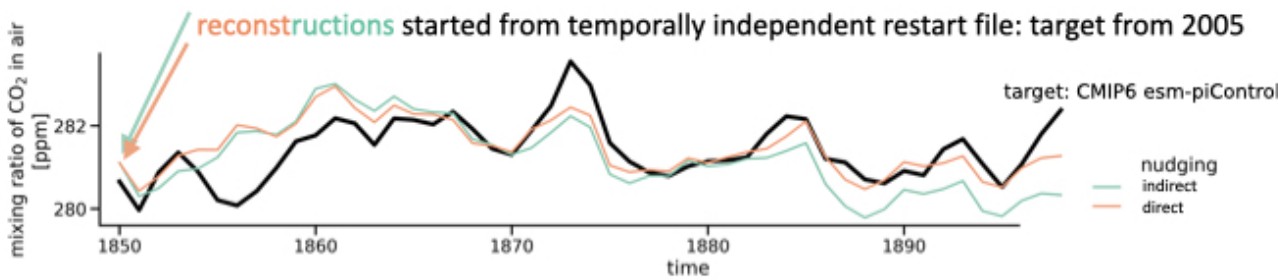

**Figure F1.** a) a) Schematic of nudging with relaxation constant. b) Schematic of reconstruction towards a target, where reconstructions are started from temporally independent restart files from the same simulation but 155 years later in time, i.e., 2005.

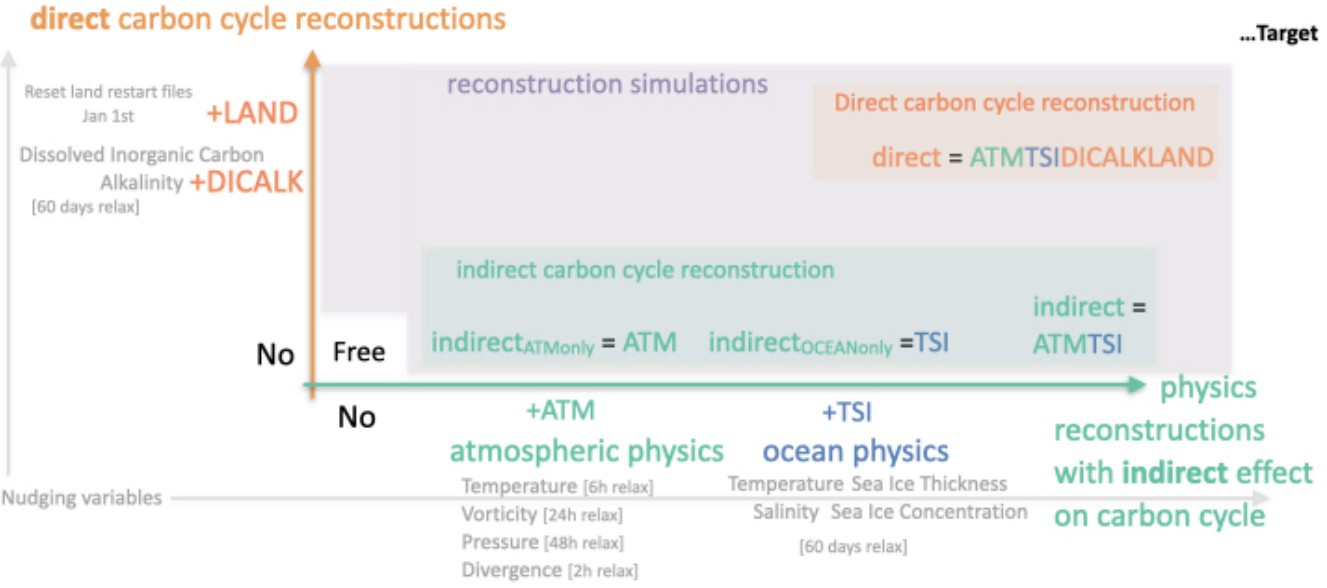

**Figure F2.** Schematic overview of perfect-model target reconstruction simulations showing which variables are reconstructed in which simulations.

## Appendix G: Climatology

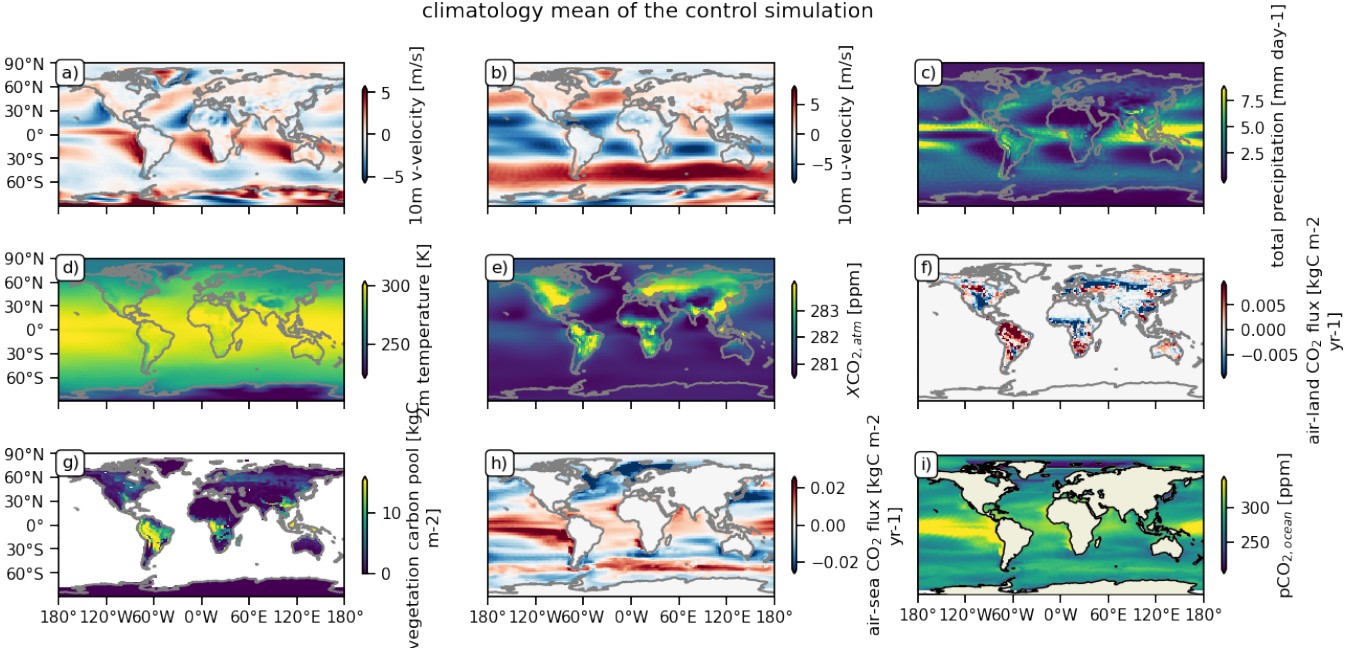

**Figure G1.** Mean climatology of the control simulations for all variables.

## Appendix H: Predictive skill LAI

We used LAI in a previous internal iteration of the paper, but chose to replace LAI with cVeg. In our model JSBACH, LAI depends on climate, it is not a carbon variable. Therefore we did not want to use this variable in the manuscript. However, there is an indirect link from LAI to air-land $CO_2$ flux, because LAI reflects droughts and the soil physics. A recent analysis focused on process-based understanding of land carbon predictability using JSBACH indicates that soil moisture as well as soil carbon storage influence the air-land $CO_2$ flux at most (Dunkl et al., 2021, under review).

*Author contributions.* A.S. and T.I. conceived the study, A.S. performed the simulations and analysis, created the figures and drafted the manuscript. T.I., I.D., H.L. and V.B. contributed in manuscript editing and provided feedback.

*Competing interests.* The authors declare no competing interests.

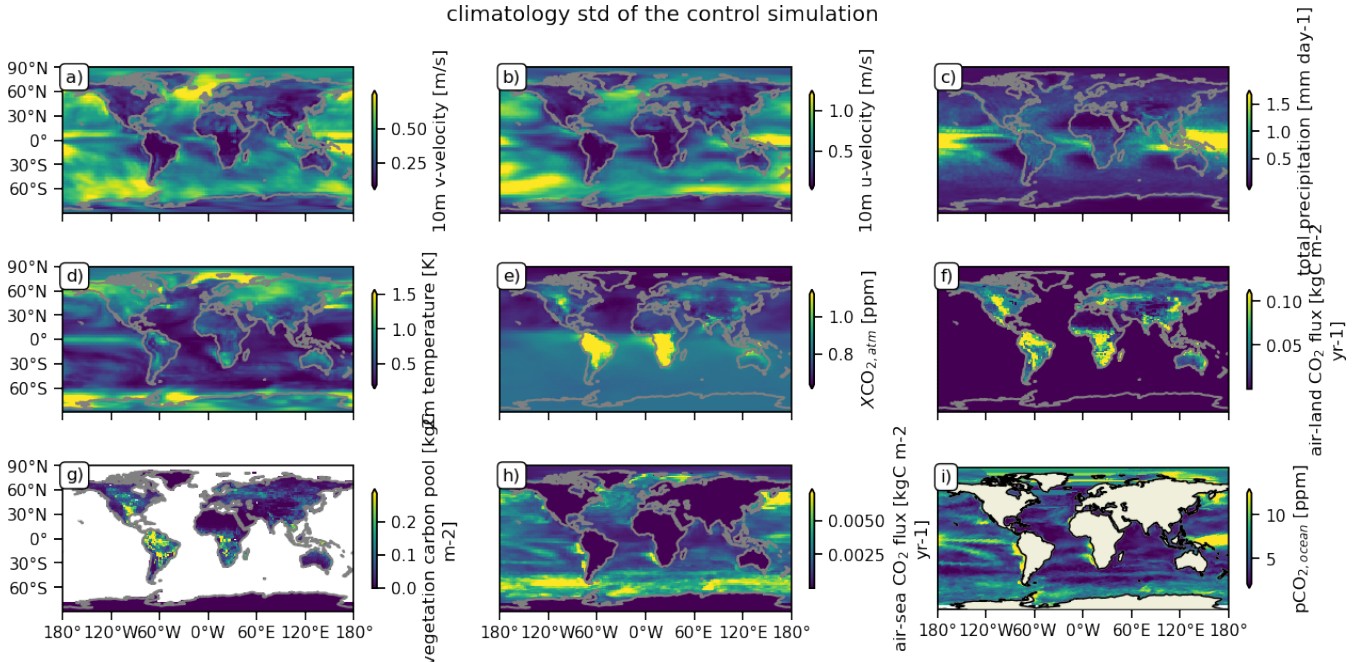

**Figure G2.** Temporal internal variability expressed as temporal standard deviation from the control simulations for all variables.

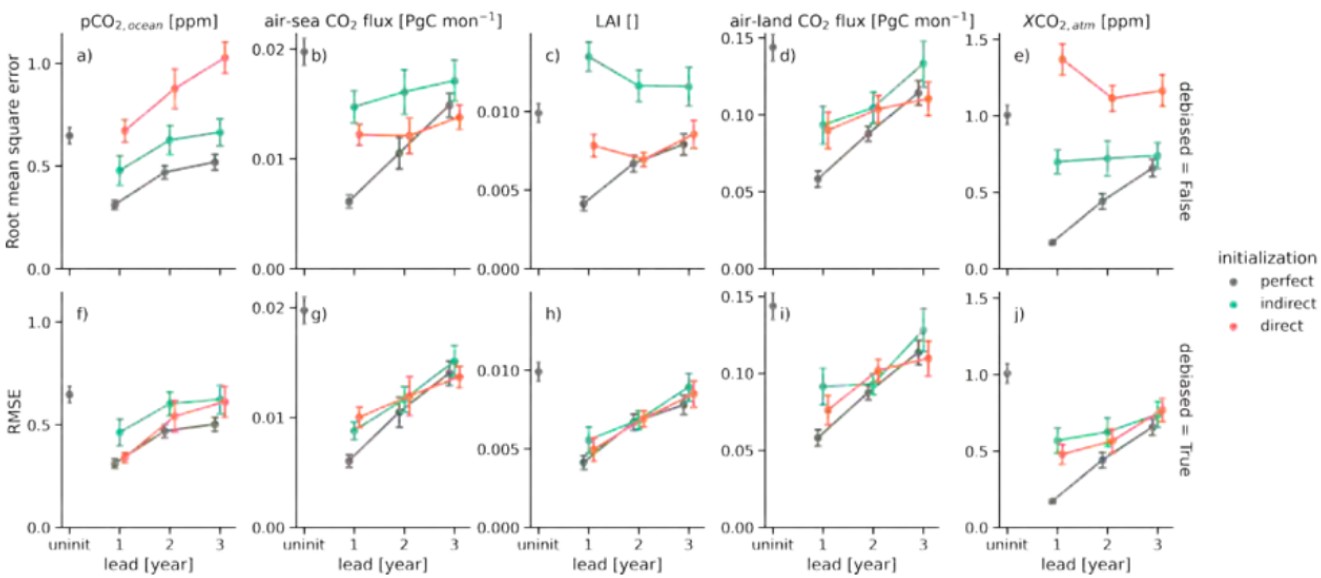

**Figure H1.** As Fig. 6 but with leaf area index (LAI) instead of carbon vegetation pools (cVeg).

*Acknowledgements.* We acknowledge funding from European Union's Horizon 2020 research and innovation programme under grant agreement No 821003 "Climate-Carbon Interactions in the Current Century (4C)" , No 820989 "COMFORT" and No 641816 "CRESCENDO". Simulations were performed at the German Climate Computing Center (DKRZ). We thank Jürgen Bader for internal review.

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
