# Peer review of "Trivial improvements of predictive skill due to direct reconstruction of the global carbon cycle"

_Earth System Dynamics, 2021_

## Author Comment (AC1)

We thank both Reviewers for their time in revisiting this manuscript. Please find our point-by-point response (normal font) to all comments (*italics*) below.

**Reviewer #1: John Dunne**

**General comments:**

*The manuscript "Trivial improvements of predictive skill due to direct reconstruction of global carbon cycle" by Spring et al. describe results from reconstructed simulations nudging a model simulation to itself for atmospheric physics, ocean physics, and carbon cycle as well as perfect predictability experiments with the original model and these reconstructions. I like this paper very much as a detailed investigation into the limits of injesting "data" into a model.*

Thank you for your supportive assessment of our work.

*The text is extremely dense, with many concepts and model limitations all being discussed at once with a focus on biases without any description of the mean state, which made me need to read over sections over and over before I was able to put the pieces together. For example, the degredation of ITCZ and Southern Ocean winds is only clear after one puts together a mental map of the base state. It might help to have the base simulation of each parameter in Figure 1 as a new Figure 1 or provided as supplementary material.*

We will provide a climatology figure in the supplementary to enhance readability of the manuscript.

*The use of language was combersome, however, with such vague words as "indirect", "direct" and "reconstruction" are used when descriptive terms like "physically nudged" "physically nudged atmosphere" and "phsyically and biogeochemically nudged" would have worked. I am guessing that there is a literature precedent for this redirection of terms, but it made the early parts of the manuscript difficult to maintain in scope.*

We searched the literature and are not aware of a precedent, which we would happily comply with. Added Fig. 1 & 2 for further clarification. We decided against biogeochemically nudged, because we only nudge DIC and alkalinity, but no further biogeochemical variables. We plan to add the following figures for better understanding of the simulations to the supplementary.

*The discussion of Figure 2 is incomplete and extends out through Figure 5.*

We agree with the Reviewer's criticism of the current manuscript structure. In the manuscript we deal with multiple variables from the ocean, land, and atmospheric model components, analyze different spatial domains (gridded or global), and apply multiple metrics (bias, correlation, accuracy). An option would be to split the figure content into one figure per variable with all the different metrics to compare, which we find somewhat unpractical. Alternatively, what we choose, is to split the figure content into one figure per metric/method, where all variables are comparable in one figure.

In either structure, it is challenging to avoid the jumps in the discussion between the figures, because data presented in the figures is discussed and analyzed under various angles allowing to formulate our main conclusions. We

[Figure]

Figure 1: a) Schematic of nudging with relaxation constant. b) Schematic of reconstruction towards a target, where reconstructions are started from temporally independent restart files from the same simulation but 155 years later in time, i.e., 2005.

[Figure]

Figure 2: Schematic overview of perfect-model target reconstruction simulations showing which variables are reconstructed in which simulations.

believe that in a full-size manuscript, such approach is adequate. We will revise the manuscript to reduce such jumping, to consolidate the discussion of result with the purpose of enhancing the readability of the manuscript. Please note that in Fig. 2 we discuss land and ocean $CO_2$, one driver variable and the result in atmospheric $CO_2$. As we first discuss ocean in section 3.2, then land in section 3.3 and finally the atmosphere 3.4, the discussion of Fig. 2 is indeed spread out.

*The conclusions seem a bit wanting of the opportunities for future investigation. Rather than being satisfied with "We conclude that the indirect carbon cycle reconstruction serves its purpose." It would be much more productive to point out what alternatives to nudging might provide superior options for future work. It should also be noted in the conclusion that the present work does not address the potential role for structural uncertainty, and potential for ecosystems to be more complex than represented in the current model and thus needing external constraint and providing a potential advantage to "direct initialization".*

In response to this comment, we add to the text: "However, our study is designed and conducted in an idealized framework. When transferring our results into assimilation of real-world observations and its implications on predictability, structural uncertainties (model resolution in space and time) and missing ecosystem processes need to additionally be dealt with." And: "Future studies, especially those aiming to address regional marine ecosystem services, could consider a wider range of assimilation techniques and data breadth. Furthermore, more the implications of more advanced data assimilation techniques for the carbon cycle [Evensen, 1994; Han et al., 2004; Balmaseda et al., 2007; Zhang et al., 2007] should be explored."

**Technical comments:**

*p1,ln8 – "We nudge variables from this target onto arbitrary initial conditions 150 years later mimicking an assimilation simulation generating initial conditions for hindcast experiments of prediction systems" I don't understand how this process works from this description. There is also a comma missing after "later" Instead, it sounds like the authors "nudged variables towards simulations from the same run 150 years earlier" to create a reconstruction of the target dataset.*

Indeed, we nudged variables from the target simulation onto a temporally independent restart file, where we took a restart file from the same simulation but from 155 years after the nudging period. We added Fig. 1 for clarity and clarified the sentence to "We nudge variables from this target onto arbitrary initial conditions from a restart file 155 years later, mimicking an assimilation simulation".

*P1 ln12 - I don't quite understand the distinction between "direct reconstruction" and indirect reconstruction". It is not defined in the abstract.*

We will rewrite the abstract and make these definitions more clear.

*Abstract overall – I think this is the longest abstract I have ever seen, yet it only describes concepts vaguely. I recommend the authors strip out the details of internally defined distinctions and spend more time on the implications of "We find improvements in global carbon cycle predictive skill from direct reconstruction compared to indirect reconstruction. After correcting for mean bias, indirect and direct reconstruction both predict the target similarly well and only*

*moderately worse than perfect initialization after the first lead year."*

We will shorten the abstract accordingly.

*Ln41 – "where the forecast is started from" is redundant.*

Deleted.

*Ln 44 – does "indirectly" translate to "uninitialized"?*

No. "Uninitialized" would mean that the model simulation is not initialized from reconstructions/observational products. "indirectly" refers to initialised simulations in which the climate state is initialized, but the carbon cycle is not initialized directly from data. Yet it is not uninitialized like in the historical CMIP6 simulations, where climate evolution of internal variability does not necessarily follow the observed/target climate evolution. We add explanation for the terminology used in the manuscript.

*Ln 55 – This sentence is an identity "In this perfect-model target reconstruction framework, we have perfect knowledge about the ground truth and a perfect model"*

It is a definition and explicit description.

*Ln 58 – "Originally"? A reference should be provided as to the early work that is being invoked.*

Added [Schneider and Griffies, 1999] and [Meehl et al., 2009] for an overview of initialized predictions.

*Ln 60 and 61 – This appears to be describing results and conclusions of the present work. References should be provided to establish the literature context (as is done on ln 62).*

Ln 60 is now changes as introductory sentence for paragraph, more citations now added: We describe what nudging is used for [Jeuken et al., 1996] and list previous studies on the impacts of nudging on circulation of ocean-atmosphere interaction [Zhu and Kumar, 2018] and biogeochemistry [Toggweiler et al., 1989].

*Ln 65 – How do you know about these "severe consequences"? what is the citation? I know that this problem is discussed in the following, but there must be others:.*
We refer to the studies cited above and missed to include Park et al., 2018 before.

*Ln 91 – This is a strange justification. One could make the same argument for N2 or O2... presumably the reason for focusing on carbon has more to do with relevance to society. Is the question being answered why land and ocean are being treated together? If so, perhaps "We focus on the combined ocean and*

*land aspects of the carbon cycle because this allows us to explore the implications of flux predictability for atmospheric CO2 as well-mixed greenhouse gas."*

The sentence meant to strengthen why we need to address this issue in global simulations and why only looking at $CO_2$ fluxes would not suffice. Changed to "We focus on the global carbon cycle, because the land and ocean carbon cycle control the internal variability of atmospheric $CO_2$."

*Ln 244-245 – "dominated by the bias of pCO2" instead of the bias in temperature?*

$CO_2$ flux is influenced by oceanic $pCO_2$, which is influenced by oceanic temperature, circulation and biology.

*Ln 248 – The description of this figure suddenly stops without addressing the XCO2 panels.*

We discuss land and atmosphere in a later section. We had to decide: either discuss all variables first and the reader has to jump in between indirect and direct reconstructions, or discuss all reconstructions first and then variables are separated between section (our choice). We add a note where the land and atmosphere subfigures will be discussed.

*Figure 4 – it would be helpful for the reader to see the comparative lines for Indirect Atm only to compare with Figure 2 and Figure 3*

Because the manuscript is already quite long and we want to confront the direct and the indirect method, we choose to only show a comparison of indirect (atmosphere and ocean nudging) with direct (atmosphere, ocean and carbon cycle). We included indirect ATM in figures 1 and 2 to show the improvement of ocean assimilation compared to just assimilating the atmosphere, as it is done in the Global Carbon Budget. However, our primary goal is to compare the current standard of the indirect carbon cycle reconstruction by atmosphere and ocean assimilation in recent studies [Li et al., 2019; Yeager et al., 2018; Park et al., 2019].

*Ln 302 – not sure why this sentence has its own paragraph*

We tried to structure of the atmospheric $CO_2$ reconstruction subsection in a way that we first describe the indirect method and then the direct carbon cycle reconstruction. This single sentence section 3.4 introduces the next two sections 3.4.1 and 3.4.2. We will merge this into few sections.

*Ln 347 - It is only here, after Figure 5 is presented, that I get to find why Figure 2o looks so much like Figure 2m. If I understand correctly, it is a coincidence – Figure 2m is high because the surface temperature is high, while Figure 2o is high because the land releases CO2 over the course of the year do to the climate mismatch. A statement to this effect near Ln 248 before moving on to Figure 3 would help orient the reader.*

We agree that this explanation for the similar plots in 2m,o is missing. We

add to section 3.4.2, where we describe direct carbon cycle reconstruction (after previous Ln 358): "There are very similar spatial distribution of the bias for the indirect atmosphere-only and the direct carbon cycle reconstruction [Fig. 2m,o], with different underlying causes. In the indirect atmosphere-only reconstruction, the $XCO_2$ bias is high [Fig. 2m], because the carbon cycle outgasses due to higher global temperatures due to nudging [Fig. 1d]. However, in the direct reconstruction the bias is high [Fig. 2o], because the carbon stocks from land and ocean are reconstructed without mass conservation. Here, the temperature bias is smaller than in the indirect atmospheric nudging, because the ocean temperature nudging stabilizes the global heat content. "

*Ln 407 – "but below the initialized" is unclear, is "but drifts slightly below the initialized value over the course of the simulation" intended?*

Indeed unclear. We change to ", but the error stays below the direct initialization error, which suffers from the bias in the direct reconstruction simulation."

*Ln 422 – "For a real-world application, our direct land carbon reconstruction method cannot be used." I would disagree with this statement and should change "cannot" to "should not". The easiest form of data assimilation for land would be to simply over-write the vegetation biomass periodically from a satellite product, something very similar in principle to what is being done here. I think the more interesting question that is answered here is why that is a bad idea. I think this is a point very much worth making as satellite products become more diverse and land initialization approaches are considered.*

We agree that our statement was a bit too strong. We change to "should not". We append discussing: "In practice satellite products of carbon cycle variables could be assimilated into the model periodically or at each time step. However, just strong interference with the model will likely result in strong drifts, especially in dependent variables. For useful real-world applications of land carbon cycle assimilation, sequential [Evensen, 1994; Balmaseda et al., 2007; Zhang et al., 2007] or variational data assimilation techniques [Han et al., 2004] could be used for initialization. But still the problem of data availability for the reforecast period remains."

*Ln 424 – This conclusion appears to be the crux of the paper – that the nudging technique introduces such large biases in climate mean state as to make the "direct" approach incompatible with the original model. I am not an expert on physical data assimilation, but isn't that the reason that ensemble Kalman filter is used rather than nudging? Would one expect these other techniques that do not shift the ITCZ or dampen Southern Ocean winds to also find a "trivial" role for BGC initialization?.*

We agree that the biases after nudging are substantial, which is the case for all reconstruction simulations, which all lead to their own peculiar differences as Reviewer notes above. We want to note that we are not aware of other studies trying reconstruct a model climatology with that same model target. Usually, observations are reconstructed and therefore there is no clear target to compare

to.

The nudging method is one of the simplest reconstruction methods. Because it can be quite strong and abrupt, ensemble Kalman filter is replacing the method over time. We add in section 5: "Haney reconstruction is the simplest approach to data assimilation allowing little flexibility to the model. Many centers are now transitioning towards the ensemble Kalman filter data assimilation, which allows more variability [Park et al., 2019; Brune and Baehr, 2020]. Applying such techniques to the carbon cycle may lead to better reconstructions."

*457 - Rather than being satisfied with "We conclude that the indirect carbon cycle reconstruction serves its purpose." It would be much more productive to point out what alternatives to nudging might provide superior options for future work. It should also be noted in the conclusion that the present work does not address the potential role for structural uncertainty to provide an advantage to "direct initialization"*

Inspired by the above comments, we already added more recent methods applicable to carbon cycle reconstruction. We clarify: "We conclude that the indirect carbon cycle reconstruction serves its purpose of reconstructing variation in the global carbon cycle."

We agree that we only use one model to reconstruct to itself. Therefore we do not have any structural uncertainty other than the reconstruction method itself or processes missing in our framework. We agree when reconstructing the real world, our model lacks processes and resolution contributing to, our framework lacks structural uncertainty. These points are added to section 5.

**References**

Balmaseda, M. A., D. Dee, A. Vidard, and D. L. T. Anderson (2007). "A Multivariate Treatment of Bias for Sequential Data Assimilation: Application to the Tropical Oceans". en. In: *Quarterly Journal of the Royal Meteorological Society* 133.622, pp. 167–179. DOI: `10/czgj3m`.

Brune, Sebastian and Johanna Baehr (2020). "Preserving the Coupled Atmosphere–Ocean Feedback in Initializations of Decadal Climate Predictions". en. In: *WIREs Climate Change* 11.3, e637. DOI: `10/ghtnt8`.

Estella-Perez, Victor, Juliette Mignot, Eric Guilyardi, Didier Swingedouw, and Gilles Reverdin (2020). "Advances in Reconstructing the AMOC Using Sea Surface Observations of Salinity". en. In: *Climate Dynamics* 55.3, pp. 975–992. DOI: `10/gkbhsh`.

Evensen, Geir (1994). "Sequential Data Assimilation with a Nonlinear Quasi-Geostrophic Model Using Monte Carlo Methods to Forecast Error Statistics". en. In: *Journal of Geophysical Research: Oceans* 99.C5, pp. 10143–10162. DOI: `10/fpjxh8`.

Fransner, Filippa, François Counillon, Ingo Bethke, Jerry Tjiputra, Annette Samuelsen, Aleksi Nummelin, and Are Olsen (2020). "Ocean Biogeochemical Predictions—Initialization and Limits of Predictability". English. In: *Frontiers in Marine Science* 7. DOI: `10/gg22rr`.

Friedlingstein, Pierre et al. (2020). "Global Carbon Budget 2020". English. In: *Earth System Science Data* 12.4, pp. 3269–3340. DOI: `10/ghn75s`.

Han, Guijun, Jiang Zhu, and Guangqing Zhou (2004). "Salinity Estimation Using the $T$ - $S$ Relation in the Context of Variational Data Assimilation: SALINITY ESTIMATION". en. In: *Journal of Geophysical Research: Oceans* 109.C3. DOI: `10/b34qr8`.

Ilyina, T. et al. (2021). "Predictable Variations of the Carbon Sinks and Atmospheric CO2 Growth in a Multi-Model Framework". en. In: *Geophysical Research Letters* 48.6, e2020GL090695. DOI: `10/ghsn7h`.

Jeuken, A. B. M., P. C. Siegmund, L. C. Heijboer, J. Feichter, and L. Bengtsson (1996). "On the Potential of Assimilating Meteorological Analyses in a Global Climate Model for the Purpose of Model Validation". en. In: *Journal of Geophysical Research: Atmospheres* 101.D12, pp. 16939–16950. DOI: `10/d64x9q`.

Li, H., T. Ilyina, W. A. Müller, and P. Landschützer (2019). "Predicting the Variable Ocean Carbon Sink". en. In: *Science Advances* 5.4, eaav6471. DOI: `10/gf4fxm`.

Lovenduski, N. S., S. G. Yeager, K. Lindsay, and M. C. Long (2019). "Predicting Near-Term Variability in Ocean Carbon Uptake". In: *Earth System Dynamics* 10.1, pp. 45–57. DOI: `10/gfvxkc`.

Meehl, Gerald A. et al. (2009). "Decadal Prediction: Can It Be Skillful?" In: *Bulletin of the American Meteorological Society* 90.10, pp. 1467–1486. DOI: `10/dpsjbp`.

Park, Jong-Yeon, Charles A. Stock, Xiaosong Yang, John P. Dunne, Anthony Rosati, Jasmin John, and Shaoqing Zhang (2018). "Modeling Global Ocean Biogeochemistry With Physical Data Assimilation: A Pragmatic Solution to the Equatorial Instability". In: *Journal of Advances in Modeling Earth Systems* 10.3, pp. 891–906. DOI: `10/gddxmt`.

Park, Jong-Yeon, Charles A. Stock, John P. Dunne, Xiaosong Yang, and Anthony Rosati (2019). "Seasonal to Multiannual Marine Ecosystem Prediction with a Global Earth System Model". en. In: *Science* 365.6450, pp. 284–288. DOI: 10/gf7fbj.

Pohlmann, Holger, Johann H. Jungclaus, Armin Köhl, Detlef Stammer, and Jochem Marotzke (2009). "Initializing Decadal Climate Predictions with the GECCO Oceanic Synthesis: Effects on the North Atlantic". In: *Journal of Climate* 22.14, pp. 3926–3938. DOI: 10/cdvhcr.

Rast, Sebastian, Renate Brokopf, Monika Esch, Veronika Gayler, Ingo Kirchner, Luis Kornblueh, Andreas Rhodin, and Uwe Schulzweida (2012). *User Manual for ECHAM6*. Tech. rep. Hamburg. URL: https://icdc.cen.uni-hamburg.de/fileadmin/user_upload/icdc_Dokumente/ECHAM/echam6_userguide.pdf.

Ruprich-Robert, Yohan et al. (2021). "Impacts of Atlantic Multidecadal Variability on the Tropical Pacific: A Multi-Model Study". en. In: *npj Climate and Atmospheric Science* 4.1, pp. 1–11. DOI: 10/gkb6tb.

Sarmiento, Jorge Louis and Nicolas Gruber (2006). *Ocean Biogeochemical Dynamics*. eng. Princeton, NJ: Princeton Univ. Press. URL: https://press.princeton.edu/books/hardcover/9780691017075/ocean-biogeochemical-dynamics.

Schimel, David S. (1995). "Terrestrial Ecosystems and the Carbon Cycle". en. In: *Global Change Biology* 1.1, pp. 77–91. DOI: 10/dw5kbg.

Schneider, Tapio and Stephen M. Griffies (1999). "A Conceptual Framework for Predictability Studies". en. In: *Journal of Climate* 12.10, pp. 3133–3155. DOI: 10/cf6zsg.

Séférian, Roland, Laurent Bopp, Marion Gehlen, Didier Swingedouw, Juliette Mignot, Eric Guilyardi, and Jérôme Servonnat (2014). "Multiyear Predictability of Tropical Marine Productivity". en. In: *Proceedings of the National Academy of Sciences* 111.32, pp. 11646–11651. DOI: 10/f6cgs3.

Servonnat, Jérôme, Juliette Mignot, Eric Guilyardi, Didier Swingedouw, Roland Séférian, and Sonia Labetoulle (2015). "Reconstructing the Subsurface Ocean Decadal Variability Using Surface Nudging in a Perfect Model Framework". en. In: *Climate Dynamics* 44.1-2, pp. 315–338. DOI: 10/f6v7kq.

Spring, Aaron and Tatiana Ilyina (2020). "Predictability Horizons in the Global Carbon Cycle Inferred From a Perfect-Model Framework". In: *Geophysical Research Letters* 47.9, e2019GL085311. DOI: 10/ggtbv2.

Toggweiler, J. R., K. Dixon, and K. Bryan (1989). "Simulations of Radiocarbon in a Coarse-Resolution World Ocean Model: 1. Steady State Prebomb Distributions". en. In: *Journal of Geophysical Research: Oceans* 94.C6, pp. 8217–8242. DOI: 10/ffvkfj.

Yeager, S. G. et al. (2018). "Predicting Near-Term Changes in the Earth System: A Large Ensemble of Initialized Decadal Prediction Simulations Using the Community Earth System Model". In: *Bulletin of the American Meteorological Society*. DOI: 10/gddfcs.

Zhang, S., M. J. Harrison, A. Rosati, and A. Wittenberg (2007). "System Design and Evaluation of Coupled Ensemble Data Assimilation for Global Oceanic Climate Studies". EN. In: *Monthly Weather Review* 135.10, pp. 3541–3564. DOI: 10/dkvk78.

Zhu, Jieshun and Arun Kumar (2018). "Influence of Surface Nudging on Climatological Mean and ENSO Feedbacks in a Coupled Model". en. In: *Climate Dynamics* 50.1, pp. 571–586. DOI: 10/gcwrz4.

---

## Author Comment (AC2)

We thank both Reviewers for their time in revisiting this manuscript. Please find our point-by-point response (normal font) to all comments (*italics*) below.

**1 Reviewer #2: anonymous**

*The authors have performed a set of perfect model experiments to investigate the impact of the nudging various physical (indirect reconstruction) and biogeochemical variables (direct reconstruction) on the reconstruction of the ocean, land and atmospheric carbon cycle. Further, they look into how this reconstruction impacts the predictive skill of the carbon cycle. They found that nudging of physical state variables reconstructs the carbon cycle well, and that an additional nudging of biogeochemical state variables only gives marginal improvement, and sometimes even deteriorates the reconstruction. Also for the predictive skill they do not find any substantial improvements of directly reconstructing the carbon cycle.*

*This manuscript is an important contribution to the research on carbon reconstruction and prediction. I do not know of any perfect model studies investigating biogeochemical reconstruction and the importance of direct versus indirect reconstruction. Further, their results on the predictability add confidence to the results of another perfect model study that showed that the biogeochemical initial conditions play a minor role for the predictability of ocean biogeochemistry.*

Thank you for your positive evaluation of our work.

*However, there are some improvements that can be done before a potential publication: I have one question mark regarding the presentation of your results. For the carbon reconstruction, why do you present the results from the atmospheric nudging and the atm+ocean+ice nudging for the indirect simulations? I do not know of any prediction systems that nudge atmospheric variables only (I may be wrong). It is rather the opposite. It is standard practice to nudge ocean variables, and then there might be nudging of atmospheric variables as an "add-on". I therefore do not see what the scientific community gains from your experiment with atmospheric nudging only. It is quite intuitive that you cannot reconstruct the ocean carbon cycle by assimilating only atmospheric data. I think that the manuscript would greatly improve if you presented the simulation where you nudge the ocean only, and then the atm+ocean+ice simulation. If you want to go into details you could even have one ocean simulation, one ocean+ice and one ocean+ice+atmosphere. In that way you could see what additional skill you could gain when nudging sea ice and atmospheric variables for carbon reconstructions.*

Historically, predictions originate from meteorology/weather predictions where only the atmosphere has been nudged. It is why decadal or seasonal predictions emerged into a separate discipline that on longer time scales a coupled ocean and atmosphere, GCMs are needed. Yet, another step forward was to perform ocean assimilation. Not all prediction systems do this, see [Ilyina et al., 2021, Table S1]. For instance, CESM-DPLE [Yeager et al., 2018] uses atmospheric reanalysis data to reconstruct the ocean, and IPSL-CM6A-LR [Estella-Perez et al., 2020] only assimilates some sea surface variables.

In our opinion nudging just the atmosphere is similar to what ocean-only (atmosphere forced) simulations do, which are like the Global Carbon Budget [Friedlingstein et al., 2020], where ocean-only and land-only simulations are forced with reanalysis. (The main difference is that nudging the atmosphere still allows for some ocean/land-atmosphere feedbacks perturbed by nudging, whereas forced simulations do not have dynamic feedbacks.) We also include the atmosphere only initialisation simulations in figure 1 and 2, as they help to understand that the strong biases are due to the atmospheric nudging. Brune and Baehr [2020] review reconstruction methods ranging from an initialization of atmospheric variability only to full-field nudging of both atmosphere and ocean.

We have already performed such ocean only reconstruction simulations. However, the figures are already quite packed with subfigures. Maybe just concentrating on indirect and direct in the manuscript, while keeping indirect ATM only and indirect OCEAN only to the supplementary information is an option.

*The discussion of the results needs some work. Specifically, the manuscript lacks a deeper discussion on advantages/disadvantages with you various nudging schemes, and the reasons behind (i.e. how does it affect the physics/biogeochemistry, regional differences). For example, why does the indirect reconstruction result in higher correlations in the subtropical ocean compared to the extratropical ocean? It is already done to some extent, but it can be improved. Moreover, you need to put your results more in context to what is currently done in reconstruction/prediction research. For example, at the moment it is not clear from the text what you want to show with the different indirect reconstruction schemes, and what knowledge we can gain from it. Overall you need to connect your results better to the literature. For example, you refer to the Servonnat et al., 2015 and Fransner et al., 2020 papers in the introduction, but you do not put your results into context with them in the discussion. How does your results compare to other studies with biogeochemical reconstructions/predictions?*

We want to show that indirect reconstruction is quite good and that improving the reconstructions by injesting carbon cycle data is tricky. We missed these references in the conclusions and will add: "This result is similar to Fransner et al. [2020], who find that ocean carbon cycle initial conditions matter much less than physical ocean initial conditions for annual carbon cycle predictions." We mostly use the method of Servonnat et al. [2015], but do not evaluate how surface nudging penetrates into the subsurface ocean as they do.

*The structure of the paper can be improved, specifically, the number of sections and subsections can be drastically reduced. See suggestions under Specific comments.*

We thought that more sections and subsections improve readability, but we are open to the suggestions.

*Work needs to be done on the language/formulations and flow on sentences. I have put some suggestions under "Technical Notes" below.*

**Contents**

Figure 3: Table of contents of the pre-print.

Agreed. Thank you for such constructive feedback.

**Specific comments:**

*I suggest you merge sections 2.2, 2.3 and 2.5 to one. You could use subsubsections for the different kinds of simulations. Similarly, I would suggest you to merge sections 2.4 and 2.6. You can furthermore reduce the number of subsections in section 3. I would simply merge 3.2.1 and 3.2.2 under 3.2, 3.3.1 and 3.3.2 under 3.3 and 3.4.1 and 3.4.2 under 3.4.*

We will merge these sections.

*Furthermore, I would suggest you to make sections 3 and 4 to subsections in a section "Results" or "Results and DIscussion". (It is difficult for me to understand whether sections 3 and 4 are supposed to be only Results or Results and Discussion)*

Both sections 3 and 4 include elements of results and discussions, but the separation supports the storyline of the paper in a more coherent way (in our

view) that the two guiding questions posed (1. how direct/indirect reconstruction matter and 2. what are the implications for the global C-cycle) are addressed in a consequential manner. We hope the merges from the previous question will also make our rationale more clear.

*Why don't you do any indirect reconstruction of the land physics? You should explain this.*

We explain in section 2.3 that "there is no nudging module available in the land surface model JSBACH." The current structure of JSBACH code is not flexible enough to allow frequent rewriting of physical variable fields, such as soil moisture or temperature, with external data. This feature will be included into the new JSBACH version which is currently in a testing phase in the new ICON model. We decided against implementing nudging routine into JSBACH version from scratch and use instead replacement of restart file with all land variables every year. With our approach, however, we make a first-order estimate of the impact of nudging on the air-land CO2 fluxes.

*Under section 3.4 you write that you will start by examining figure 4, but then you only mention it briefly in the end of the section.*

We meant to say we start analysing the variables driving globally averaged atmospheric $CO_2$ with an emphasis on global. Figure 4 shows the timeseries of atmospheric $CO_2$, which is why we set the figure reference. The title of 3.4 clearly indicates that this section is about reconstruction skill, so is the paragraph describing Fig. 5 in detail.

*Why are you looking into seasonal timescales when it comes to atmospheric CO2 (figure 5), while you are looking into 10 year chunks for the land and the ocean? I would focus on one time-scale to be consistent, but maybe I'm missing something.*

Figure 5 looks into all months, because we are aware of the limits of the hard reset land reconstruction, which by design results in very good January skill. The bias in the land variables quickly establishes after a few months. This is a rather technical detail. We will simplify Fig. 5 by omitting the monthly data and thereby reducing the scope of the manuscript.

*The manuscript would gain a lot of you would discuss the regional patterns in our results in more detail. For example, for the ocean we see large differences in the reconstruction skill in the tropics compared to the extra-tropics, why is this?*

We tried to explain some of the numerous regional features. We mention that "Reconstruction over the oceans is more successful in the tropics than in the extra-tropics, where the Northern and Southern Hemisphere mid-latitude westerlies have low, but still significant correlation. (L.212)". We cannot fully explain this here and mention that response of the tropics vs. extra-tropics is a challenging topic identified previously [Ruprich-Robert et al., 2021]. We are unable to solve this here and can only call for more attention. We suggest

that perhaps the pace-making experiments, such as those suggested within the DCPP model intercomparison could be a framework to address this some of the regional challenges.

*In your ACC plots, the ACC that is significantly different than internal variability, is found in areas of lower correlation. Why is this? Shouldn't the significant result be related to higher correlation?*

Our description in the figure was misleading. Stippling is shown, where reconstruction skill is worse (lower ACC) than an upper 95% threshold of resampled internal variability. This is changed to "i.e. the reconstruction is not significantly different than a resampled internal variability threshold"

*Why have you chosen RMSE as a measure of skill for your predictions, while you use RMSE and ACC for your reconstructions? Did you try also ACC for your predictions?*

We tried ACC for predictions, but as shown in [Spring and Ilyina, 2020] RMSE is more useful for $CO_2$ fluxes when aiming for atmospheric $CO_2$, which are the accumulated $CO_2$ fluxes. Furthermore ACC is independent of bias, but here we have strong biases.

*Section 4.2: if the cVeg is not predictable, what is it then that yields the predictability in the air-land CO2 flux? If you cannot give an answer to this you should at least discuss it. (for the ocean you are looking into the oceanic pCO2 that is an important driver for the air-sea co2 flux.)*

Good point. LAI is not shown, but predictable as air-land $CO_2$ flux. We'll add this to the text. We used LAI in a previous internal iteration of the paper, but chose to replace LAI with cVeg. In our model JSBACH, LAI depends on climate, it is not a carbon variable. Therefore we did not want to use this variable in the manuscript. However, there is an indirect link from LAI to air-land $CO_2$ flux, because LAI reflects droughts and the soil physics. Recent analysis focused on process-based understanding of land carbon predictability using JSBACH indicates that soil moisture as well as soil carbon storage influence the air-land CO2 flux at most.

**Technical Notes:**

**Abstract**

: The abstract is long and heavy to read. Try to shorten it down and make the text more fluent. Some suggestions:

We will shorten the abstract and are thankful for the recommendations.

*The first sentences are not capturing the reader., and need reformulation. Maybe you could start by saying that state-of-the art climate prediction systems now include a carbon component. Then you shortly explain that while there is assimilation of physical state variables, this is not the case for the biogeochemistry.*

[Figure]

Figure 4: As Fig. 6 but with LAI instead of cVeg.

We will shorten the abstract and are thankful for this recommendation.

*line 8 : Why have you chosen the word "target"? It is quite abstract, could you choose another word? If you want to keep it, maybe just a reformulation of the text would help.*

We are using target in this paper after Servonnat et al. [2015] uses this term.

*Lines 10-15: This is very much into technical detail, and I suggest to remove most of this from the abstract.*

We will shorten the abstract and are thankful for this recommendation.

*I would suggest not to go into ACC's and RMSE's in the abstract, and instead give your interpretations of your results in words. If you want to keep it, only mention it for the most important results.*

Agreed.

**Introduction**

*Move lines 47-56 to the end of the introduction*

We choose to first ask the research questions and then integrate and refine them afterwards into the existing literature. We are considering moving them near the end of the intro.

*line 55: the word perfect- model target reconstruction framework is very long. Can you make this shorter? Can't you just write "perfect model framework", and then you describe it in more detail in your methodology?*

We tried to make the distinction between section 3 reconstruction of the target and section 4 prediction of that target.

*Lines 74-77: This part, which is on ocean physics, comes in the middle of your discussion on initialization of the carbon cycle (in between the references to Li et al., 2016, 2019 , Seferian et al., 2018, Lovenduski, et al., 2019 and Fransner et al., 2020). I suggest moving it to the end of the introduction where you discuss your approach. If I understood it right you got inspired from this study?*

Exactly, reading Servonnat et al. [2015] inspired this study. This paragraph (Ln 70 - 92) is about reconstruction methods and not about ocean carbon cycle initialization. We just mention these land and ocean carbon predictability studies to of the assumed weakness of these indirect reconstruction studies. We first describe the two methods found in the literature and then define what exactly we do.

*Lines 74-75: You should make it clear already in this first sentence that Servonnat et al is a perfect model study.*

Agree.

*Lines 75-77: This sentence is difficult to understand. There are several reasons behind; i) "target reconstruction approach" is quite abstract, and needs clarification, i.e. why is the method of Servonnat et al., 2015 called like this? Why does it allow to directly assess the quality of reconstructed initial conditions? The last part "which is useful and practical to know for forecaster issuing a forecast" also need a reformulation, or can be removed completely .*

Exactly. Servonnat et al. [2015] define and use this term. This approach allows to compare the quality of reconstructed initial conditions with the target immediately, because the target and the reconstruction are both available. We think such an immediate (dont want to say direct) comparison is valuable, because a forecaster should know how close the the initial conditions are to the truth. In the real-world we dont know the truth, but in a perfect-model framework we do.

*lines 84-85: in which way is this more theoretical?*

We mean more theoretical here in the way explained above. It is useful theoretical knowledge from a twin experiment that the initial conditions of the ocean carbon matter less than from the physical ocean. We try to generate practical knowledge showing how good or bad the reconstructed initial conditions are.

**Methods**

*lines 103-107: you repeat the reference Mauritsen et al., 2019 three times. Maybe this is not needed if you talk about the same model all the time*

Agreed.

*lines 115-117: here comes the explanation for the "perfect-model target re-construction framework" that I was looking for in the introduction. I would suggest to move these first lines there. Alternatively, you only use the phrasing "perfect model framework" before this section. Then the exact methodology and details behind are described here. In that case you can keep these lines here. Please consider making two sentences out of this one. it is very long.*

Separated into three sentences. I will refer to "perfect-model framework" before only.

*Lines 117-118: how can the restart file for year 2005 come from the per-industrial control simulation?*

We are in a pre-industrial control simulation and tried to indicate by "model year" that we are not taking the literal 2005 state. Furthermore, perfect-model framework makes it clear that this is not about observed climate. We mentioned the model year 2005 for reproducibility.

*Line 119: you have already described the model setup in the previous section, and do not have to write about is again here.*

The previous reference was about the model version. This reference here is about the simulation setup.

*lines 126-128 + equation: here comes the explanation for the use of the word "target" that I was looking for earlier. I would suggest moving this after the first sentence in section 2.2.*

We also add Fig. 1 to the supplementary. We prefer to keep this subsection as is.

*lines 132-134: The observational data is not needed at each model time step, but at the time scale of relaxation, right?*

Correct.

*line 136: I would suggest to write "a shorter relaxation time scale" instead of a "a stronger nudging strength"*

Point taken.

*line 138-139: you need to clarify why you nudge the logarithm of the surface pressure. Is the reference to Pohlmann et al related to this, or to the nudging of all variables?*

[Pohlmann et al., 2009] refers to all variables. We cannot justify why the logarithm of pressure is taken. It seems standard practice. Since [Jeuken et al., 1996] until today [Rast et al., 2012].

*line 139: I would suggest you to briefly mention the 63 spherical harmonics in the model description.*

Added.

*line143: I don't understand the use of the word "only" here. Should it be nudging the atmosphere only?*

Removed "Only".

*line 144: consider merging the parentheses that are coincident.*

Done.

*line 156: consider removing this first sentence, the information in it is basically repeated in the next sentence.*

Agreed.

*Lines 157-160:To avoid repetition, I would remove the "over 10 year windows" in the first sentence. In the second sentence you can then explain that you do the calculations over ten year windows, and why you do it. Moreover, I would move the second sentence to be after the explanation of your skill-metrics.*

Implemented.

**Section 3**

*figure 1: the letters in subplots g-l are barely visible, consider changing the color to white. Check this also for the other figures.*

We will do so in the re-submission.

*section 3.1: why are you looking at these physical variables, specifically? A short explanation would be good.*

"We identify atmospheric circulation represented by winds and resulting precipitation and temperature to be descriptive for the impact of circulation on the carbon cycle [Schimel, 1995; Sarmiento and Gruber, 2006]."

*lines 211-212: The ACC is not significantly better in most grid cells from what it looks like in Figure 1.*

We had to go over all figures mentioning the stippling (showing not better tracking performance than internal variability resampled threshold) in manuscript. For Fig. 1 g-l most areas show better tracking performance than internal variability and no stippling.

*line 238: "the state variable of the ocean carbon sink surface ocean pCO2" needs reformulation*

Reformulated to "we focus on air-sea $CO_2$ flux and surface oceanic $pCO_2$ as the state variable of the ocean carbon sink, which is the oceanic driver of air-sea $CO_2$ flux."

*Lines 246-248: Here it would be interesting if you discussed the results more in detail and put it into context with the results in 3.1. For example, how come that the atmospheric nudging only improves the reconstruction in the tropics?*

Added "The additional reconstruction of the physical ocean [Fig. 1e,f] enables largely a correlation above 0.7 [Fig. 3b,e]." The tropics remains not better reconstructed than the threshold. We mislead you with the figure caption about the stippling. Adding "Only the carbon cycle in the tropical oceans remain difficult to reconstruct due to the strong biases in atmospheric circulation [Fig. 1a,b,c]."

*Line 260: the biases in the direct reconstruction does not look larger than the ones in the indirect reconstruction for pCO2 in Figure 2 ?*

Reconstructing DIC and alkalinity in the ocean reduce oceanic $pCO_2$ bias and improve correlation. Changed to "Section 3.2 shows how well indirect and direct reconstruction of the ocean carbon cycle work overall. While the direct reconstruction has slightly larger biases in air-sea $CO_2$ flux, direct reconstruction also brings higher correlation."

*section 3.4.1: here you are again describing the effect of the reconstruction on the land and ocean carbon cycle as you did in sections 3.2 and 3.3 . I would remove it from here and only discuss the effect on the atmospheric CO2. The description of the land and the ocean carbon cycles should be done in the previous sections.*

We deliberately focus here in section 3.4.1 on the global air-land and air-sea $CO_2$ flux when we are discussing the effects of reconstruction on global atmospheric $CO_2$, whereas we discuss the spatial distributions in section 3.2 and 3.3.

**Section 4**

*figure 6: Use the same y-label for the two rows in the left hand side, i.e. either RMSE or root mean squared error*

Ok.

*line 389: add flux after "global air-sea CO2"*

Added.

*373: you have to describe how you construct this perfectly initialized ensemble in the methods. You write that it is started from perfect initial conditions. I assumed that you applied some perturbation to these restart files?*

Perturbation as described in 2.5 where we also add "The perfectly initialized ensembles are started from the target initial conditions without any previous reconstruction simulation."

**Section 5**

*Line 435: shouldn't it be "reconstruction of the physical ocean fields"?*

Agreed. We use initialization and reconstruction synonymous here. Reconstruction fits better.

*Line 435-436: It is quite expected that you cannot reconstruct the ocean carbon cycle by nuding the atmosphere only no? You need to discuss this in more detail. Also please put your results into context of what is currently done in state-of-the-art climate prediction models, and other studies showing that the ocean carbon cycle/biogeochemistry can be reconstructed by nudging physics only (i.e Li et al., 2016, Seferian et al., 2014, Park et al., 2018 ...).*

In our opinion the Global Carbon Budget simulations aim to do so. Added "For the ocean carbon cycle, the reconstruction of the physical ocean fields is critical to reconstruct carbon cycle initial conditions, which explain why current state-of-the-art carbon cycle prediction systems have skill despite not initializing the ocean carbon cycle with ocean carbon cycle observations [Séférian et al., 2014; Park et al., 2018; Li et al., 2019; Lovenduski et al., 2019]."

*Lines 451-460: you need to put these results into context with the Fransner et al., 2020 study.*

Added "This means that oceanic carbon cycle initial conditions are much less important than physical ocean initial conditions for oceanic carbon cycle predictions, which confirms the findings of [Fransner et al., 2020]."

**References**

Balmaseda, M. A., D. Dee, A. Vidard, and D. L. T. Anderson (2007). "A Multivariate Treatment of Bias for Sequential Data Assimilation: Application to the Tropical Oceans". en. In: *Quarterly Journal of the Royal Meteorological Society* 133.622, pp. 167–179. DOI: `10/czgj3m`.

Brune, Sebastian and Johanna Baehr (2020). "Preserving the Coupled Atmosphere–Ocean Feedback in Initializations of Decadal Climate Predictions". en. In: *WIREs Climate Change* 11.3, e637. DOI: `10/ghtnt8`.

Estella-Perez, Victor, Juliette Mignot, Eric Guilyardi, Didier Swingedouw, and Gilles Reverdin (2020). "Advances in Reconstructing the AMOC Using Sea Surface Observations of Salinity". en. In: *Climate Dynamics* 55.3, pp. 975–992. DOI: `10/gkbhsh`.

Evensen, Geir (1994). "Sequential Data Assimilation with a Nonlinear Quasi-Geostrophic Model Using Monte Carlo Methods to Forecast Error Statistics". en. In: *Journal of Geophysical Research: Oceans* 99.C5, pp. 10143–10162. DOI: `10/fpjxh8`.

Fransner, Filippa, François Counillon, Ingo Bethke, Jerry Tjiputra, Annette Samuelsen, Aleksi Nummelin, and Are Olsen (2020). "Ocean Biogeochemical Predictions—Initialization and Limits of Predictability". English. In: *Frontiers in Marine Science* 7. DOI: `10/gg22rr`.

Friedlingstein, Pierre et al. (2020). "Global Carbon Budget 2020". English. In: *Earth System Science Data* 12.4, pp. 3269–3340. DOI: `10/ghn75s`.

Han, Guijun, Jiang Zhu, and Guangqing Zhou (2004). "Salinity Estimation Using the $T$ - $S$ Relation in the Context of Variational Data Assimilation: SALINITY ESTIMATION". en. In: *Journal of Geophysical Research: Oceans* 109.C3. DOI: `10/b34qr8`.

Ilyina, T. et al. (2021). "Predictable Variations of the Carbon Sinks and Atmospheric CO2 Growth in a Multi-Model Framework". en. In: *Geophysical Research Letters* 48.6, e2020GL090695. DOI: `10/ghsn7h`.

Jeuken, A. B. M., P. C. Siegmund, L. C. Heijboer, J. Feichter, and L. Bengtsson (1996). "On the Potential of Assimilating Meteorological Analyses in a Global Climate Model for the Purpose of Model Validation". en. In: *Journal of Geophysical Research: Atmospheres* 101.D12, pp. 16939–16950. DOI: `10/d64x9q`.

Li, H., T. Ilyina, W. A. Müller, and P. Landschützer (2019). "Predicting the Variable Ocean Carbon Sink". en. In: *Science Advances* 5.4, eaav6471. DOI: `10/gf4fxm`.

Lovenduski, N. S., S. G. Yeager, K. Lindsay, and M. C. Long (2019). "Predicting Near-Term Variability in Ocean Carbon Uptake". In: *Earth System Dynamics* 10.1, pp. 45–57. DOI: `10/gfvxkc`.

Meehl, Gerald A. et al. (2009). "Decadal Prediction: Can It Be Skillful?" In: *Bulletin of the American Meteorological Society* 90.10, pp. 1467–1486. DOI: `10/dpsjbp`.

Park, Jong-Yeon, Charles A. Stock, Xiaosong Yang, John P. Dunne, Anthony Rosati, Jasmin John, and Shaoqing Zhang (2018). "Modeling Global Ocean Biogeochemistry With Physical Data Assimilation: A Pragmatic Solution to the Equatorial Instability". In: *Journal of Advances in Modeling Earth Systems* 10.3, pp. 891–906. DOI: `10/gddxmt`.

Park, Jong-Yeon, Charles A. Stock, John P. Dunne, Xiaosong Yang, and Anthony Rosati (2019). "Seasonal to Multiannual Marine Ecosystem Prediction with a Global Earth System Model". en. In: *Science* 365.6450, pp. 284–288. DOI: `10/gf7fbj`.

Pohlmann, Holger, Johann H. Jungclaus, Armin Köhl, Detlef Stammer, and Jochem Marotzke (2009). "Initializing Decadal Climate Predictions with the GECCO Oceanic Synthesis: Effects on the North Atlantic". In: *Journal of Climate* 22.14, pp. 3926–3938. DOI: `10/cdvhcr`.

Rast, Sebastian, Renate Brokopf, Monika Esch, Veronika Gayler, Ingo Kirchner, Luis Kornblueh, Andreas Rhodin, and Uwe Schulzweida (2012). *User Manual for ECHAM6*. Tech. rep. Hamburg. URL: `https://icdc.cen.uni-hamburg.de/fileadmin/user_upload/icdc_Dokumente/ECHAM/echam6_userguide.pdf`.

Ruprich-Robert, Yohan et al. (2021). "Impacts of Atlantic Multidecadal Variability on the Tropical Pacific: A Multi-Model Study". en. In: *npj Climate and Atmospheric Science* 4.1, pp. 1–11. DOI: `10/gkb6tb`.

Sarmiento, Jorge Louis and Nicolas Gruber (2006). *Ocean Biogeochemical Dynamics*. eng. Princeton, NJ: Princeton Univ. Press. URL: `https://press.princeton.edu/books/hardcover/9780691017075/ocean-biogeochemical-dynamics`.

Schimel, David S. (1995). "Terrestrial Ecosystems and the Carbon Cycle". en. In: *Global Change Biology* 1.1, pp. 77–91. DOI: `10/dw5kbg`.

Schneider, Tapio and Stephen M. Griffies (1999). "A Conceptual Framework for Predictability Studies". en. In: *Journal of Climate* 12.10, pp. 3133–3155. DOI: `10/cf6zsg`.

Séférian, Roland, Laurent Bopp, Marion Gehlen, Didier Swingedouw, Juliette Mignot, Eric Guilyardi, and Jérôme Servonnat (2014). "Multiyear Predictability of Tropical Marine Productivity". en. In: *Proceedings of the National Academy of Sciences* 111.32, pp. 11646–11651. DOI: `10/f6cgs3`.

Servonnat, Jérôme, Juliette Mignot, Eric Guilyardi, Didier Swingedouw, Roland Séférian, and Sonia Labetoulle (2015). "Reconstructing the Subsurface Ocean Decadal Variability Using Surface Nudging in a Perfect Model Framework". en. In: *Climate Dynamics* 44.1-2, pp. 315–338. DOI: `10/f6v7kq`.

Spring, Aaron and Tatiana Ilyina (2020). "Predictability Horizons in the Global Carbon Cycle Inferred From a Perfect-Model Framework". In: *Geophysical Research Letters* 47.9, e2019GL085311. DOI: `10/ggtbv2`.

Toggweiler, J. R., K. Dixon, and K. Bryan (1989). "Simulations of Radiocarbon in a Coarse-Resolution World Ocean Model: 1. Steady State Prebomb Distributions". en. In: *Journal of Geophysical Research: Oceans* 94.C6, pp. 8217–8242. DOI: `10/ffvkfj`.

Yeager, S. G. et al. (2018). "Predicting Near-Term Changes in the Earth System: A Large Ensemble of Initialized Decadal Prediction Simulations Using the Community Earth System Model". In: *Bulletin of the American Meteorological Society*. DOI: `10/gddfcs`.

Zhang, S., M. J. Harrison, A. Rosati, and A. Wittenberg (2007). "System Design and Evaluation of Coupled Ensemble Data Assimilation for Global Oceanic Climate Studies". EN. In: *Monthly Weather Review* 135.10, pp. 3541–3564. DOI: `10/dkvk78`.

Zhu, Jieshun and Arun Kumar (2018). "Influence of Surface Nudging on Climatological Mean and ENSO Feedbacks in a Coupled Model". en. In: *Climate Dynamics* 50.1, pp. 571–586. DOI: 10/gcwrz4.